# Visual-O1: Understanding Ambiguous Instructions via Multi-modal Multi-turn Chain-of-Thoughts Reasoning

**Minheng Ni**[1,2]**, Yutao Fan**[2]**, Lei Zhang**[1]✉ **& Wangmeng Zuo**[2,3]✉
[1]Department of Computing, Hong Kong Polytechnic University
`minheng.ni@connect.polyu.hk`  `cslzhang@comp.polyu.edu.hk`
[2]Faculty of Computing, Harbin Institute of Technology  [3]Pengcheng Laboratory
`fanyutao@stu.hit.edu.cn`  `wmzuo@hit.edu.cn`

## Abstract

As large-scale models evolve, language instructions are increasingly utilized in multi-modal tasks. Due to human language habits, these instructions often contain ambiguities in real-world scenarios, necessitating the integration of visual context or common sense for accurate interpretation. However, even highly intelligent large models exhibit observable performance limitations on ambiguous instructions, where weak reasoning abilities of disambiguation can lead to catastrophic errors. To address this issue, this paper proposes Visual-O1, a multi-modal multi-turn chain-of-thought reasoning framework. It simulates human multi-modal multi-turn reasoning, providing instantial experience for highly intelligent models or empirical experience for generally intelligent models to understand ambiguous instructions. Unlike traditional methods that require models to possess high intelligence to understand long texts or perform lengthy complex reasoning, our framework does not notably increase computational overhead and is more general and effective, even for generally intelligent models. Experiments show that our method not only enhances the performance of models of different intelligence levels on ambiguous instructions but also improves their performance on general datasets. Our work highlights the potential of artificial intelligence to work like humans in real-world scenarios with uncertainty and ambiguity. We release our data and code at https://github.com/kodenii/Visual-O1.

## 1 Introduction

With the advancement of deep learning, increasing attention is being paid to multi-modal scenarios that are more relevant to reality, such as visual question answering (VQA) (Antol et al., 2015; Zhou et al., 2020; Teney et al., 2018), referring image segmentation (RIS) (Lai et al., 2023; Ni et al., 2023; Yang et al., 2023), and vision-and-language navigation (VLN) (Li & Bansal, 2023; Hong et al., 2021; Feng et al., 2022). In recent years, more manipulable language instructions are gradually being introduced into these tasks to better align with human interaction habits. Combined with large-scale language models (OpenAI, 2024; Chen et al., 2023a; Liu et al., 2024), artificial intelligence (AI) is beginning to use language instructions closer to real-world scenarios to perform tasks, broadening the range of AI applications..

As shown in Figure 1, due to the inherent ambiguity of natural language and the excellent analytical abilities of humans, the language instructions used by humans often contain vagueness and ambiguity. Additionally, human language and vision are closely related and often require combining visual context or common sense to accurately understand their meanings. Therefore, ambiguous instructions, which are common in reality, differ from meticulously designed accurate instructions (Antol et al., 2015), and learning to understand them directly becomes challenging in the absence of corresponding task data.

Recently, chain-of-thoughts (CoT) reasoning has greatly enhanced the understanding and analytical capabilities of high-intelligent large models, such as GPT-4o. However, its application to scenar-

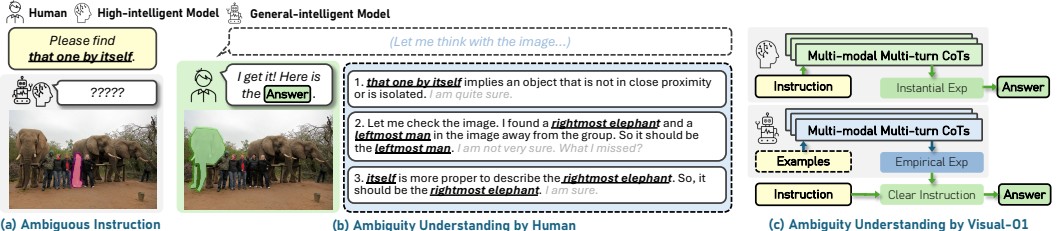

Figure 1: **Understanding ambiguous instruction.** The AI model may not be able to execute instructions normally when encountering ambiguous instructions. However, humans can usually correctly analyze the actual meaning of ambiguous instructions by combining visual context and can accurately interpret ambiguous instructions. Based on this, we propose VISUAL-O1, which simulates human multi-modal multi-turn reasoning to gain instantial (for high-intelligent models) or empirical (for general-intelligent models) experience in order to understand ambiguous instructions.

ios involving multi-modal ambiguity understanding has yet to be explored. Additionally, general-intelligence models used in multi-modal tasks, like LLAVA, often lack the data and parameter capacity to perform chain reasoning and analysis, making it difficult to apply CoT methods to enhance their understanding of ambiguous instructions.

To address these challenges, we propose VISUAL-O1, a multi-modal multi-turn chain-of-thought reasoning method that simulates human multi-modal multi-turn reasoning. VISUAL-O1 builds instance-specific experience during inference for high-intelligence models or creates general experience for any ambiguous instructions through one-time optimization with several examples for general-intelligence models. This helps models correctly understand ambiguous instructions and synthesize the final answer.

We also construct a dataset containing various types of ambiguous instructions, including *ellipsis*, *colloquialism*, *subjectivity*, *relativity*, and *other*, to validate performance across different multi-modal scenarios. Experiments show that our method improves the performance of models with varying intelligence levels on the ambiguous instruction dataset and enhances their performance on general datasets. Ablation studies demonstrate that VISUAL-O1 can be easily applied to different multi-modal models and tasks.

Our contributions are three-fold:

- We reveal the capabilities of multi-modal models in analyzing and executing ambiguous instructions by setting up a novel benchmark for understanding ambiguous instructions in various multi-modal tasks.

- We propose VISUAL-O1, a multi-modal multi-turn chain-of-thought reasoning method, to build instantial or empirical experience for high-intelligent or general-intelligent models, enabling them to correctly understand ambiguous instructions.

- Experimental results show that our method improves the performance of models with varying intelligence levels on ambiguous instruction datasets and enhances their performance on general datasets.

## 2 RELATED WORK

**Language Instruction Understanding in Multi-modal Tasks**   Language instructions were often used as conditions in multi-modal tasks, serving as an essential medium for user-AI interaction. They had been applied in a wide range of highly reality-related tasks (Reed et al., 2016; Bigham et al., 2010; Zellers et al., 2019). Traditionally, language instructions were precise and unambiguous. Although some works had managed to understand and execute complex language instructions by combining large multi-modal models (LMMs) (Lai et al., 2023; Yang et al., 2023), they still lacked a practical understanding of ambiguous instructions. Recently, a few works had noticed the presence of ambiguity in language within specific multi-modal tasks. In image classification, WAFFLECLIP (Roth et al., 2023) and FUDD (Esfandiarpoor & Bach, 2023) had pointed out the issue of polysemy

in classification texts, using LLMs to generate commonalities or differences within categories to enhance image classification tasks. In visual question answering, Prasad et al. (2023) proposed REPHRASE to repeatedly utilize LLMs to mine image information through manually pre-designed prompts to enhance language understanding.

However, existing methods required models to engage in extensive interactions with samples based on predefined rules to mine information, requiring extremely long inference times and relying on specific tasks and models, making it difficult to generalize to various tasks and models with different intelligence levels.

**Complex Reasoning with Large Multi-modal Models**   In recent years, large language models (LLMs) have expanded into multi-modal scenarios (OpenAI, 2024; Liu et al., 2024; Lai et al., 2023), showcasing the potential of large multi-modal models (LMMs) in handling multi-modal tasks. However, challenges remain in understanding complex language instructions within visual contexts in multi-modal tasks. Recently, methods to enhance the reasoning capabilities of large multi-modal models have attracted some research attention. On one hand, some works have proposed training-based methods. Dai et al. (2024) proposed a two-stage training method that aligned pre-trained models with images and texts, enhancing the capabilities of LMMs. Similarly, Chen et al. (2023b); Wang et al. (2024) achieved impressive results in multi-modal tasks using data generated by GPT. On the other hand, some works have proposed non-training methods, such as the chain-of-thoughts (CoT) (Wei et al., 2022; Yao et al., 2024) approach, which enhances model understanding by simulating human reasoning.

However, existing methods either relied on a large amount of real or model-generated data to optimize model parameters, which is challenging to scale cost-effectively to any task, or they required the model to have strong reasoning abilities, making it challenging to general-intelligent models.

## 3   VISUAL-O1: MULTI-MODAL MULTI-TURN CHAIN-OF-THOUGHTS REASONING FRAMEWORK

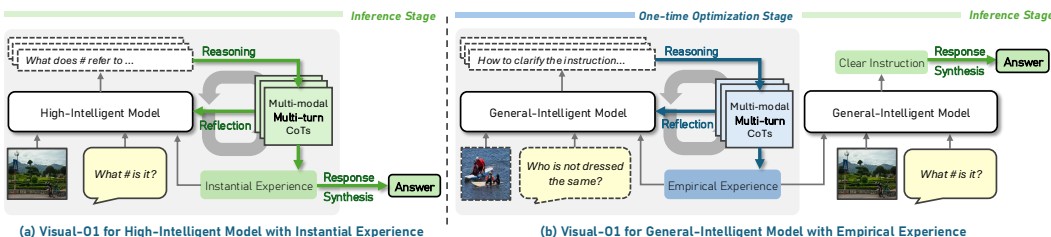

Figure 2: **The overview of VISUAL-O1.** VISUAL-O1 introduces multi-modal multi-turn chain-of-thoughts to understand ambiguity with (a) instantial experience for high-intelligent models to generate the correct answer directly, and (b) empirical experience for general-intelligent models to transform ambiguous instructions into clear instructions and then generate the correct answer. Instantial and empirical experience develops during inference and one-time optimization stage.

### 3.1   OVERVIEW OF PROPOSED FRAMEWORK

Given a task that requires both a visual context $\mathbf{x}_v$ and an ambiguous instruction $\mathbf{x}_a$, it can be performed by a multi-modal model $f$. The result of the task is denoted as $\mathbf{y} = f(\mathbf{x}_a, \mathbf{x}_v \mid \theta_f)$, where $\theta_f$ represents the parameters of the task model[1]. Due to the ambiguous or unclear information contained in the text instruction $\mathbf{x}_a$, the performance of the model $f$ can be largely affected. If this model is a general-intelligent model, this can have catastrophic effects due to its weaker reasoning abilities, making it nearly impossible for the model to produce the correct result. Even for high-intelligent large multi-modal models, we still observe performance limitations (see Tables 1 and 2 for more details).

---

[1]Since we do not optimize $\theta_f$, we omit it in the following text for brevity

To address this issue, as shown in Figure 2, we propose VISUAL-O1, a multi-modal multi-turn chain-of-thoughts reasoning framework. VISUAL-O1 aims to help models build experience in disambiguating ambiguous instructions, enabling them to correctly understand the instructions and produce the right answers. Unlike traditional chain-of-thought methods that require models to possess high intelligence for understanding long texts and performing complex reasoning, our framework is more general and effective even for general-intelligent models.

Specifically, for high-intelligent models, VISUAL-O1 establishes sample-level disambiguation experience during inference by leveraging the ambiguous instructions themselves. For general-intelligent models, VISUAL-O1 builds empirical disambiguation experience during one-time optimization using several examples. VISUAL-O1 synthesizes responses by either summarizing the sample-level experience into the final answer or using the empirical experience to transform ambiguous instructions into clear instructions. This clear instruction is then combined with the original ambiguous instruction to summarize the final answer. Despite the slight differences in Visual-O1 for high-intelligent and general-intelligent models, their structures are completely consistent. Therefore, we can formulate them within a unified framework.

## 3.2 REASONING AND REFLECTION

Given a multi-modal model $f$, by utilizing the visual context and logical reasoning, we can supplement vague information, enabling the model to understand the precise meaning of the ambiguous instruction. To achieve this, we perform a reasoning step under the guidance of a pre-defined prompt $p_{\text{rsn}}$ to attempt to understand the ambiguous instruction:

$$\mathbf{x}_{\text{rsn}} = f(\mathbf{x}_a, \mathbf{x}_v \mid p_{\text{rsn}}; \mathcal{A}), \tag{1}$$

where $\mathbf{x}_{\text{rsn}}$ is the reasoning result, and $\mathcal{A}$ is a special prompt that contains the disambiguation experience. Our goal is to obtain this special prompt $\mathcal{A}$ to help $f$ understand the ambiguities.

Assuming we complete one reasoning step, next, the model reflects on the reasoning result $\mathbf{x}_{\text{rsn}}$. Assuming $p_{\text{rfl}}$ is the predefined prompt, we obtain the reflection result $\mathbf{x}_{\text{rfl}}$ by:

$$\mathbf{x}_{\text{rfl}} = f(\mathcal{A}, \mathbf{x}_{\text{rsn}} \mid p_{\text{rfl}}). \tag{2}$$

By utilizing reflection, we summarize the issues in the reasoning process to improve the disambiguation experience $\mathcal{A}$. By repeatedly performing the reasoning-reflection process, we gradually refine the reasoning results, transforming it into an iterative form. We rewrite the reasoning and reflection steps in an iterative form as follows:

$$\mathbf{x}_{\text{rsn}}^{(i)} = f(\mathbf{x}_a, \mathbf{x}_v \mid p_{\text{rsn}}; \mathcal{A}^{(i)}), \tag{3}$$

$$\mathbf{x}_{\text{rfl}}^{(i)} = f(\mathcal{A}^{(i)}, \mathbf{x}_{\text{rsn}}^{(i)} \mid p_{\text{rfl}}), \tag{4}$$

where $i$ is the $i$-th iteration. Assuming our budget is $N$, we directly apply the iterations formed by Eq. (3-4) and an update function of $\mathcal{A}^{(i)}$ by $N$ times to obtain the final $\mathcal{A} := \mathcal{A}^{(N)}$. Next, we introduce the instantial and empirical experience of $\mathcal{A}$ and their updating mechanism in Visual-O1 for different intelligence levels of the model.

**Reasoning and Reflection for Instantial Experience** For high-intelligent models, in each iteration, VISUAL-O1 continues to analyze the ambiguous instruction based on the previous disambiguation process, then reflects on its analysis and generates feedback. These analyses and feedback form the reasoning process for the entire problem. Therefore, for the $i$-th iteration, we update $\mathcal{A}_{\text{ins}}$ by aggregating all information:

$$\mathcal{A}_{\text{ins}}^{(i+1)} = \mathcal{A}_{\text{ins}}^{(i)} \oplus \mathbf{x}_{\text{rsn}}^{(i)} \oplus \mathbf{x}_{\text{rfl}}^{(i)}, \tag{5}$$

where $\oplus$ is the concatenation function. Obviously, this process can directly work on the inference stage. Assuming our budget is $N_{\text{ins}}$, we directly apply the iterations formed by Eq. (3-5) $N_{\text{ins}}$ times during the inference stage to obtain a complete disambiguation process.

**Reasoning and Reflection for Empirical Experience** For general-intelligent models, in each iteration, VISUAL-O1 uses the existing experience to continue attempting to transform ambiguous

instructions into clear ones, then reflects on its conversion process and updates the experience. Due to the limited understanding capabilities of general intelligent models for long texts, we only retain the new reflection information as the update for $\mathcal{A}_{\text{emp}}$:

$$\mathcal{A}_{\text{emp}}^{(i+1)} = \mathbf{x}_{\text{rfl}}^{(i)}. \tag{6}$$

Similar to the process of a high-intelligent model, we apply the iterations formed by Eq. (3-4, 6) $N_{\text{emp}}$ times to obtain the final disambiguation experience. The difference is that we change one sample in each iteration, which means a total of $N_{\text{emp}}$ samples, to prevent a learning experience that is specific to certain samples rather than a generalizable experience. Fortunately, we only need to obtain $\mathcal{A}_{\text{emp}}$ once in the one-stage optimization stage; for the same task, we no longer need to repeatedly obtain $\mathcal{A}_{\text{emp}}$.

### 3.3 Response Synthesis

Although experience $\mathcal{A}$ is not the final answer, it contains the model's understanding of the instructions and is important information for solving the problem. Next, we obtain the final answer $\mathbf{y}$ through response synthesis.

**Response Synthesis by Instantial Experience** For high-intelligent models, $\mathcal{A}_{\text{ins}}$ already encompasses all the reasoning, and the model possesses strong comprehension abilities. Therefore, we simply summarize the entire reasoning process with a pre-defined response synthesis prompt $p_{\text{syn}}$ to directly obtain the final answer:

$$\mathbf{y} = f(\mathcal{A}_{\text{ins}} \mid p_{\text{syn}}), \tag{7}$$

**Response Synthesis by Empirical Experience** For general-intelligent models, considering their limited reasoning capabilities, we first use the disambiguation experience $\mathcal{A}_{\text{emp}}$ learned by the model during the one-time optimization process to transform the ambiguous instruction $\mathbf{x}_a$ into a clear instruction $\mathbf{x}_c$ based on the visual context $\mathbf{x}_v$:

$$\mathbf{x}_c = f(\mathbf{x}_a, \mathbf{x}_v \mid \mathcal{A}_{\text{emp}}). \tag{8}$$

Then, we utilize the pre-defined response synthesis prompt $p_{\text{syn}}$, allowing the model to summarize the original ambiguous instruction and the clear instruction together to obtain the final answer:

$$\mathbf{y} = f(\mathbf{x}_c, \mathbf{x}_a, \mathbf{x}_v \mid p_{\text{syn}}). \tag{9}$$

Finally, regardless of the model's level of intelligence, we obtain a final answer $\mathbf{y}$ that well-understands the ambiguous instruction $\mathbf{x}_a$ based on visual context $\mathbf{x}_v$.

### 3.4 Implementation Details

We run experiments on 10 random seeds to obtain average results. We set budgets $N_{\text{ins}}$ and $N_{\text{emp}}$ to 10 and 3 for general-intelligent and high-intelligent models respectively. Please note that due to the difficulty of performing reflection with general-intelligent models, in practice, we temporarily use a high-intelligent model at this step. Since the reflection for general-intelligent models is only conducted during a one-time optimization, our final inference still relies solely on the general-intelligent model itself without introducing additional models or computational overhead.

For more details of implementation, please refer to the **Appendix** A.

## 4 Experiments

### 4.1 Experimental Setup

To verify the effectiveness of our method, we conduct a series of experiments. We apply VISUAL-O1 to the state-of-the-art high-intelligence model and general-intelligence model on two typical multi-modal tasks: referring image segmentation (RIS) and visual question answering (VQA). All comparisons are divided into ambiguous instructions and general instructions to comprehensively evaluate the models' performance on ambiguous and non-ambiguous instructions.

Table 1: **Overall results on RIS.** VISUAL-O1 outperforms other methods and achieves notable improvements on both high-intelligent and general-intelligent models. We report the average results on different random seeds. The improvement is statistically significant with $p < 0.01$ under $t$-test.

| MODEL | AMBIGUOUS | | GENERAL | |
|---|---|---|---|---|
| | GIoU | CIoU | GIoU | CIoU |
| LISA (LAI ET AL., 2023) | 0.0237 | 0.0272 | 0.4654 | 0.4721 |
| SoM (YANG ET AL., 2023) | 0.0752 | 0.1158 | 0.3507 | 0.4154 |
| CHAIN-OF-THOUGHTS (WEI ET AL., 2022) | 0.0746 | 0.0982 | 0.4183 | 0.4326 |
| FUDD (ESFANDIARPOOR & BACH, 2023) | 0.0718 | 0.0928 | 0.3416 | 0.3534 |
| VISUAL-O1 (LISA) | **0.1131** | **0.1215** | **0.4738** | **0.4980** |
| VISUAL-O1 (SoM) | **0.1304** | **0.1756** | **0.3686** | **0.4508** |

Table 2: **Overall results on VQA.** VISUAL-O1 outperforms other methods and achieves notable improvements on both ambigous and general datasets. We report the average results on different random seeds. The improvement is statistically significant with $p < 0.01$ under $t$-test.

| MODEL | AMBIGUOUS | | GENERAL | |
|---|---|---|---|---|
| | ACC | BLEU-1 | ACC | BLEU-1 |
| LLAVA (LIU ET AL., 2024) | 8.58 | 0.4760 | 54.19 | 0.7194 |
| GPT-4O (OPENAI, 2024) | 14.10 | 0.5960 | 61.25 | 0.8192 |
| CHAIN-OF-THOUGHTS (WEI ET AL., 2022) | 15.03 | 0.3372 | 39.78 | 0.5113 |
| FUDD (ESFANDIARPOOR & BACH, 2023) | 19.18 | 0.4844 | 48.37 | 0.6454 |
| VISUAL-O1 (LLAVA) | **22.13** | **0.5095** | **57.58** | **0.7295** |
| VISUAL-O1 (GPT-4O) | **22.78** | **0.6640** | **63.14** | **0.8482** |

**Baselines and Evaluation Metrics.** We select a series of typical methods as baselines. We choose the original, unprocessed high-intelligence SoM (Yang et al., 2023) and GPT-4O (OpenAI, 2024) as well as the general-intelligence LISA (Lai et al., 2023) and LLAVA (Liu et al., 2024) as comparison models for RIS and VQA. Then, we apply CHAIN-OF-THOUGHTS (Wei et al., 2022) reasoning and FUDD (Esfandiarpoor & Bach, 2023) language explanation methods on LISA and LLAVA for further comparisons. To ensure fairness in the comparisons, we ensure that all models are under the same settings. Following previous work (Bigham et al., 2010; Kazemzadeh et al., 2014; Ni et al., 2023), for RIS, we compare gIoU and cIoU, and for VQA, we compare accuracy and BLEU-1.

**Datasets.** In addition to the general datasets, we set up extra ambiguous datasets for RIS and VQA, which only contain ambiguous instructions filtered or refined manually. We categorize the ambiguous instructions into five types based on the cause of ambiguity: *ellipsis*, *colloquialism*, *subjectivity*, *relativity*, and *other*. *Ellipsis* indicates that the ambiguity stems from omitted content; *colloquialism* indicates that the ambiguity arises from the use of informal or imprecise expressions; *subjectivity* indicates that the ambiguity is due to subjective judgments; *relativity* indicates that the ambiguity comes from implied comparisons, and *other* represents other types of ambiguity.

For more details and analysis of the dataset, please refer to the **Appendix** B and C.

## 4.2 OVERALL RESULTS

**Ambiguous instructions understanding.** As shown in the left part of Tables 1 and 2, we observe that the original model performs poorly on the distinctly different tasks of VQA and RIS. This indicates that existing methods are susceptible to ambiguous instructions and do not perform well when correct visual context needs to be integrated to understand instructions. After integrating VISUAL-O1, the performance of all tasks shows a notable improvement. This means that existing methods have the potential to understand ambiguous instructions but require proper guidance. It is worth mentioning that we observe that using VISUAL-O1, general-intelligent model, LLAVA, and LISA are capable of achieving results comparable to or even better than high-intelligent models without introducing any additional models or data during inference. This further demonstrates the value of VISUAL-O1.

Table 3: **Ambiguous instruction results on different intelligence level.** VISUAL-O1 has demonstrated powerful flexibility across different levels of intelligence.

| MODEL | VISUAL-O1 | GIOU | CIOU |
|---|---|---|---|
| LISA (LAI ET AL., 2023) | | 0.0237 | 0.0272 |
| VISUAL-O1 (LISA) | EMPIRICAL | **0.1131** | **0.1215** |
| VISUAL-O1$^+$ (LISA) | INSTANTIAL | **0.1447** | **0.1351** |
| SOM (YANG ET AL., 2023) | | 0.0237 | 0.0272 |
| VISUAL-O1$^-$ (SOM) | EMPIRICAL | **0.1143** | **0.1530** |
| VISUAL-O1 (SOM) | INSTANTIAL | **0.1304** | **0.1756** |

**General instructions understanding.** While the ability to understand ambiguous instructions is crucial, the performance of the model on general datasets is also important. As shown in the right part of Tables 1 and 2, we also find that VISUAL-O1 enhances the understanding of ambiguous instructions and greatly improves performance on general datasets. This is because improving the ability to understand instructions is beneficial even for those that do not contain ambiguity. What is even more noteworthy is that traditional COT and description-based synthesis like FUDD greatly degrade the model's performance on general datasets. This is because general-intelligent models like LLAVA find it challenging to maintain the logical chains and long-text comprehension akin to high-intelligent large models like GPT-4O. Complex reasoning tends to impair the model's understanding capabilities.

For more analysis on improvement, please refer to the **Appendix** D.

## 4.3 GENERALIZABILITY STUDIES

### 4.3.1 FLEXIBILITY TO INTELLIGENCE LEVELS

Based on the strong long-text comprehension and reasoning abilities of GPT-4V's SOM, we use VISUAL-O1 for instantial experience during the inference stage for each sample. In contrast, due to LISA's weaker reasoning abilities, we utilize VISUAL-O1 for empirical experience during the one-time optimization phase, avoiding complex reasoning for each sample during the inference stage. However, this does not mean that VISUAL-O1 cannot perform empirical experience for high-intelligence models or instantial experience for general-intelligence models. To test VISUAL-O1's adaptability to different intelligence levels, we design additional models: VISUAL-O1$^+$(LISA) using instantial experience and VISUAL-O1$^-$(SOM) using empirical experience.

As shown in Table 3, the variants of VISUAL-O1 are effective across different models. It is noteworthy that since LISA cannot comprehend and generate long texts in logical chains, we use GPT-4V as its disambiguation model during the inference stage to supplement its capabilities. Whether applied to high-intelligent models or general-intelligent models, the variants of Visual-O1 largely enhance performance on ambiguous instructions.

For more analysis on intelligence levels, please refer to the **Appendix** E.

### 4.3.2 ADAPTABILITY TO MODELS

Can VISUAL-O1 generalize to different models? We select various models on RIS to verify this, and as shown in Table 4, regardless of the model used, we observe stable improvements. This highlights the generalizability of VISUAL-O1 to different models and suggests the potential for broader applications of our method.

For more analysis on adaptability to different models and tasks, please refer to the **Appendix** F.

## 4.4 ABLATION STUDIES

### 4.4.1 EFFECTIVENESS OF COMPONENTS

To further explore the effectiveness of VISUAL-O1, we conduct ablation experiments on the general VQA data in Table 5. Reasoning and reflection are crucial for the model to correctly understand

Table 4: **Generalization studies on different models.** VISUAL-O1 is effective across different models. "-" indicates values not reported in the original paper.

| MODEL | GIoU | CIoU |
|---|---|---|
| OVSEG (LIANG ET AL., 2023) | 0.0418 | 0.0778 |
| REF-DIFF (NI ET AL., 2023) | 0.3301 | 0.3140 |
| UNIFIED-IO (LU ET AL., 2022) | – | 0.4015 |
| INSTRUCTDIFFUSION (GENG ET AL., 2023) | – | 0.3904 |
| SOM (YANG ET AL., 2023) | 0.3507 | 0.4154 |
| LISA (LAI ET AL., 2023) | 0.4654 | 0.4721 |
| GPT-4O (OPENAI, 2024) | 0.5728 | 0.5152 |
| QWEN-VL-2 (WANG ET AL., 2024) | 0.4417 | 0.3220 |
| VISUAL-O1 (SOM) | **0.3686** | **0.4508** |
| VISUAL-O1 (LISA) | **0.4738** | **0.4980** |
| VISUAL-O1 (GPT-4O) | **0.5777** | **0.5325** |
| VISUAL-O1 (QWEN-VL-2) | **0.5161** | **0.4457** |

Table 5: **Ablation studies.** Each module of VISUAL-O1 plays an indispensable role.

| MODEL | ACC | BLEU-1 |
|---|---|---|
| VISUAL-O1 (LLAVA) | **57.58** | **0.7295** |
| W/O RESPONSE SYNTHESIS | 47.59 | 0.6589 |
| W/O REASONING AND REFLECTION | 54.25 | 0.7076 |
| VISUAL-O1 (GPT-4O) | **63.14** | **0.8482** |
| W/O RESPONSE SYNTHESIS | 60.00 | 0.8254 |
| W/O REASONING AND REFLECTION | 60.60 | 0.7943 |

Table 6: **Efficiency of different budgets for VISUAL-O1.** We report the average accuracy of 10 experiments, as well as their maximum, minimum, and variance.

| VISUAL-O1 | AMBIGUOUS | | | | GENERAL | | | |
|---|---|---|---|---|---|---|---|---|
| | AVG | MAX | MIN | VAR | AVG | MAX | MIN | VAR |
| W/O BUDGET | 8.95 | 10.14 | 7.89 | 0.85 | 54.02 | 54.90 | 53.30 | 0.35 |
| W/ HUMAN EXP | 16.88 | 18.43 | 14.29 | 1.61 | 54.55 | 56.20 | 53.15 | 1.02 |
| W/ 1 BUDGET | 17.83 | 19.73 | 16.76 | 0.96 | 55.38 | 56.22 | 54.10 | 0.45 |
| W/ 2 BUDGET | 19.92 | 21.89 | 18.77 | 0.95 | 56.31 | 56.92 | 54.16 | 0.58 |
| W/ 3 BUDGET | 22.13 | 24.52 | 20.91 | 0.88 | 57.58 | 58.56 | 56.76 | 0.35 |
| W/ 4 BUDGET | 18.82 | 20.54 | 17.30 | 0.84 | 55.42 | 56.34 | 54.36 | 0.34 |
| W/ 5 BUDGET | 19.85 | 21.35 | 17.94 | 0.95 | 56.17 | 57.44 | 55.14 | 0.46 |

ambiguous instructions. The model's performance on ambiguous dataset declines noticeably without reasoning and reflection. Meanwhile, response synthesis is very important for the performance on regular data from non-ambiguous instructions, as the original instruction may contain important information, and response synthesis ensures the complete transmission of this information. Every module of VISUAL-O1 effectively improves the model's performance across different datasets.

For more explorations of the model, please refer to the **Appendix** G.

### 4.4.2 EXPLORATION OF REASONING

We design additional validation experiments to further prove the effectiveness of VISUAL-O1's reasoning process. We invite 10 volunteers to manually design disambiguation empirical experiences, which are used as benchmarks for ten tests and averaged. As shown in Table 6 shows that manually designed prompts are less effective than reasoned empirical experience, demonstrating the rationality of VISUAL-O1's design.

We also show the influence of budgets in the reasoning and reflection process of VISUAL-O1 on 10 experiments with different random seeds. However, with the reasoning and reflection process, the

Table 7: **Comparison with data augmentation.** Compared to resource-intensive data augmentation, VISUAL-O1 still achieves best performance.

| METHOD | ACC | BLEU-1 |
|---|---|---|
| LLAVA W/ AMBIGUOUS EXTRA DATA | 51.48 | 0.6863 |
| LLAVA W/ AMBIGUOUS ORIGINAL DATA | 45.08 | 0.6363 |
| LLAVA W/ NOISED ORIGINAL DATA | 39.84 | 0.5912 |
| VISUAL-O1 (LLAVA) | **57.58** | **0.7295** |

performance gradually grows and then falls again after reaching its peak. We observe that VISUAL-O1 follows a trend similar to deep learning. The performance begins to grow in the early stages of learning, but then decreases due to overfitting after reaching a particular stage.

For more explorations of reasoning process and budget $N > 3$, please refer to **Appendix** H and I.

### 4.4.3 COMPARISON OF DATA AUGMENTATION

We generate an additional 10,000 data samples for fine-tuning LLAVA, as shown in Table 7. We design three different data augmentation methods: (1) AMBIGUOUS EXTRA DATA: directly using LLAVA to synthesize an extra 10,000 sets of ambiguous instruction data; (2) AMBIGUOUS ORIGINAL DATA: rewriting 10,000 sets of original instructions into ambiguous instructions, then training the model; and (3) NOISED ORIGINAL DATA: randomly deleting or modifying 10,000 sets of original instructions, then training the model.

We find that due to the synthetic data's inability to effectively simulate instructions, there is a large performance drop in VQA. Additionally, the annotation and training costs brought by the 10,000 data samples are also very high. Therefore, VISUAL-O1 effectively solves the problem of understanding ambiguous instructions while maintaining performance on the general dataset.

For comparisons of computational cost, please refer to the **Appendix** J.

### 4.5 CASE STUDIES

How does our method help the model better understand ambiguous instructions? To delve into this issue, we visualize the results of RIS in Figure 3. We find that ambiguous instructions are common in the data and often not easily detected by humans, but this can severely affect the model's performance. After introducing VISUAL-O1, the accuracy of the instructions improves noticeably, leading to more effective outcomes.

As shown in the first example, this is a typical case of *subjectivity* ambiguity, requiring the perspective of human subjective observation for reasonable inference. The original instruction could describe most of the bears in the image, as each bear's angle has some deviation. However, when combined with the image, it is found that only one bear meets the requirements of turning and slight deviation, and VISUAL-O1 successfully infers this and provides an accurate description, enabling the task model to accurately segment the target.

In the second example, this is a typical case of *ellipsis* ambiguity, where many sentence components are omitted. The segmentation model is misled by the word "`reading`" and incorrectly segments the person in the image. Meanwhile, VISUAL-O1 accurately describes that this is a bus showing a "`reading station`" sign and even provides additional information to help the segmentation model locate the target.

In the third example, this is a typical case of *relativity* ambiguity, where comparison is used to refer to specific entities. To understand the instruction, the model must first locate the comparison object of the bottle's lighter color, which is the color of other bottles, that is, the lightest-colored bottle. When combined with VISUAL-O1, the instruction is transformed into a more accurate form, enabling the model to easily locate the target.

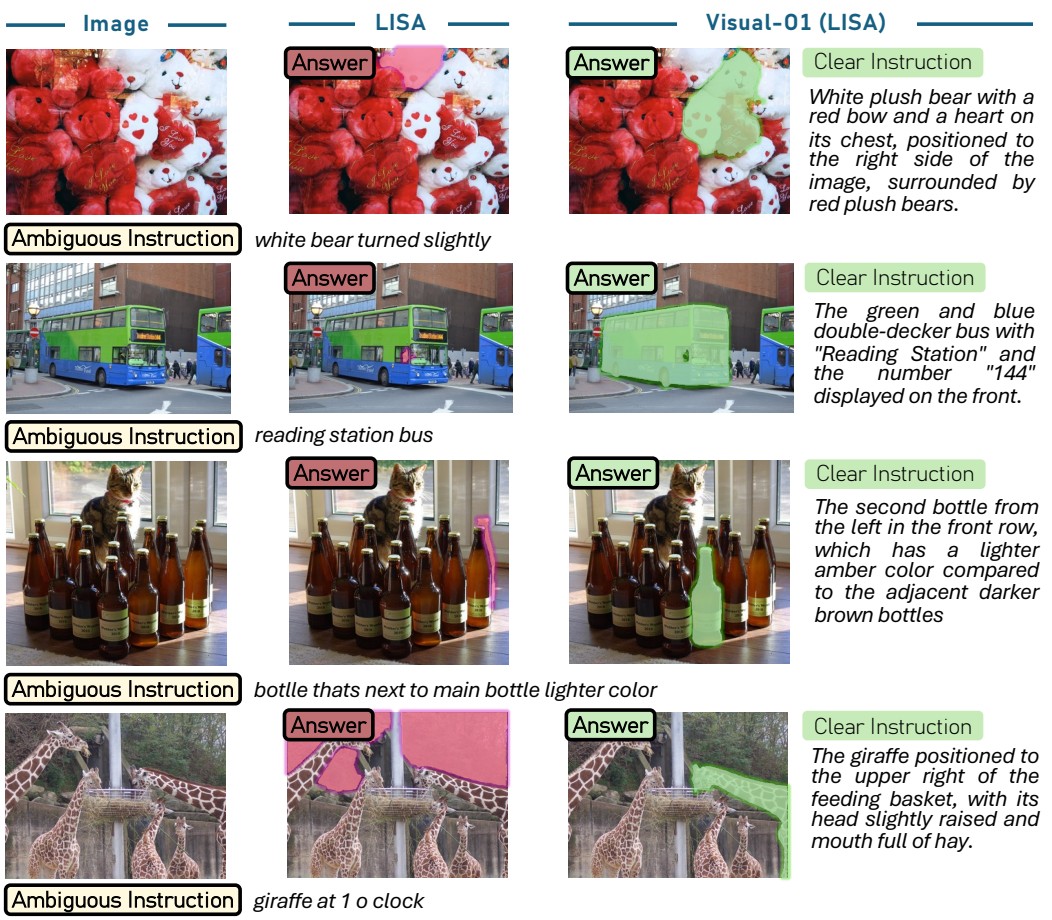

Figure 3: **Case studies on RIS.** Our approach aids the model in understanding ambiguous instructions by incorporating VISUAL-O1, which enhances the accuracy of instructions, thereby enabling more effective segmentation. Please note that ambiguous instruction may contain typos.

In the fourth example, this is a typical case of *colloquialism* ambiguity, using common positional phrases in spoken language. The model cannot understand that "1 o'clock" is a position, so it cannot select the correct object, but VISUAL-O1 transforms it into a clearer description.

For more cases, please refer to the **Appendix** K.

## 5 CONCLUSION

Even high-intelligent large models exhibited observable performance limitations on ambiguous instructions, where weak reasoning abilities of disambiguation could lead to catastrophic errors. We proposed VISUAL-O1, a multi-modal multi-turn chain-of-thoughts reasoning framework. It simulated human multi-modal multi-turn reasoning, providing instantial experience for high-intelligent models or empirical experience for general-intelligent models to understand ambiguous instructions. Unlike traditional methods that required models to possess high intelligence to understand long texts or perform lengthy complex reasoning, VISUAL-O1 did not notably increase computational overhead and was more general and effective, even for general-intelligence models. We validated our approach across various tasks and models with different intelligence levels. Experimental results demonstrated that VISUAL-O1 substantially improved the performance of models of varying intelligence on ambiguous instructions and enhanced their performance on general datasets. Our work revealed the potential of AI to operate like humans in real-world scenarios characterized by uncertainty and ambiguity.

For statements of broader impact, limitations, and reproducibility, please refer to the **Appendix** L, M, and N.

ACKNOWLEDGEMENT

This work is partially supported by the Major Key Project of PCL under Grant PCL2024A06.

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

This appendix mainly contains:

- Supplementary implementation details in Section A
- Supplementary dataset details in Section B
- Deeper analysis on dataset in Section C
- Deeper analysis on improvement in Section D
- Further generalizability verification on intelligence in Section E
- Further generalizability verification on various tasks in Section F
- Extra exploration of the model in Section G
- Extra exploration of reasoning process in Section H
- Additional ablation studies in Section I
- Additional computation studies in Section J
- Additional case studies in Section K
- Statement of broader impact in Section L
- Statement of limitations in Section M
- Statement of reproducibility in Section N

## A   SUPPLEMENTARY IMPLEMENTATION DETAILS

VISUAL-O1 uses the following prompts for high-intelligent and general-intelligent models during inference. For each prompt, the upper part is used by the high-intelligent model, while the lower part is used by the general-intelligent model.

---

**Prompt of Reasoning** $p_{\text{rsn}}$

```
You are a helpful assistant in normal conversation.
{task description}
Follow these instructions carefully:
1.  Read the given question carefully and reset counter
between budget
2.  Generate a detailed, logical step-by-step solution.
3.  Enclose each step of your solution within reasoning tags.
4.  You are allowed to use at most budget steps (starting
budget), keep track of it by counting down within tags
budget, STOP GENERATING MORE STEPS when hitting 0.  You
don't have to use all of them.

Example format:

starting budget
Content of step 1

remaining budget
Content of step 2

remaining budget
Content of step 3 or Content of some previous step

remaining budget
...

remaining budget
```

---

```
Content of final step

Description:  {ambiguous instruction}
Provide a detailed, step-by-step solution to a given
question.
---------------------------------------------------------------
{experience} {ambiguous instruction}
```

Since the reasoning process of the high-intelligent model occurs during the inference stage, to accelerate this stage, we combine all the prompts, allowing the model to complete the entire reasoning process in a single output. To clearly demonstrate, we split the prompt used by the high-intelligent model into $p_{\mathrm{rsn}}$, $p_{\mathrm{rfl}}$, and $p_{\mathrm{syn}}$.

**Prompt of Reflection** $p_{\mathrm{rfl}}$

```
Follow these instructions carefully:
5.  Do a self-reflection when you are unsure about how to
proceed; based on the self-reflection and reward, decide
whether you need to return to the previous steps.
6.  Provide a critical, honest, and subjective
self-evaluation of your reasoning process within <reflection>
and </reflection> tags.
7.  Assign a quality score to your solution as a float
between 0.0 (lowest quality) and 1.0 (highest quality),
enclosed in <reward> and </reward> tags.
8.  If the image or question is not clear enough, you need
to reflect and try to get answers from the unclear image or
question.

Example format:
<reflection> [Evaluation of the solution] </reflection>
<reward> [Float between 0.0 and 1.0] </reward>
---------------------------------------------------------------
According to the instruction you generated last time, the
annotator has rewritten {ambiguous instruction} as {clear
instruction}.  Please correct or rewrite your instruction
based on the image situation.  The image and the result of
the data annotator are only for your evaluation.  Please do
not include the specific case in the instructions.  You need
to generate the full instruction even if no change is needed.

If the annotator cannot find it, please let the annotator
guess the one with the highest probability.

Make sure the annotator only responds to the rewritten phrase
and does not include any other thing.

Instruction:  {experience}
```

Table A: **Distribution of dataset.**

| ELLIPSIS | COLLOQUIALISM | SUBJECTIVITY | RELATIVITY | OTHER |
|:---:|:---:|:---:|:---:|:---:|
| 23.3% | 27.3% | 3.3% | 29.3% | 16.7% |

---

**Prompt of Response Synthesis $p_{\text{syn}}$**

```
After completing the solution steps, reorganize and
synthesize the steps into the final answer within <answer>
and </answer> tags.  The final answer cannot be empty.
------------------------------------------------------------------
Disambiguated question:  {clear instruction}
Original question:  {ambiguous instruction}
```

---

For FUDD, the model will find the five relevant objects of ambiguous instructions in the image. Then, the model will generate the key distinguishing information for each pair of objects. The model will use this disambiguating information appending to the question to obtain a better answer. For CHAIN-OF-THOUGHTS, we use the classic prompt: "Think it step by step." and our task-modified prompt: "Analyze the question based on the image." There are slight differences in performance of two prompts and we report the higher scores.

## B SUPPLEMENTARY DATASET DETAILS

For RIS, we use the REFCOCO+ dataset (Kazemzadeh et al., 2014); for VQA, we use the VIZWIZ dataset (Gurari et al., 2018). For their ambiguous instructions, we manually screen and construct subsets of 150, 650, and 106, respectively. Specifically, we use GPT-4 for an initial screening, selecting $2,000$ sets of potentially ambiguous instructions from the original dataset. Then, we enlist multiple volunteers to manually screen for instructions that clearly contain ambiguity.

We categorize the ambiguous instructions of RIS into five distinct types based on the underlying causes of the ambiguity: *ellipsis*, *colloquialism*, *subjectivity*, *relativity*, and *other*. The distribution of ambiguities can be found in Table A. Their detailed explanations are as follows:

- *Ellipsis* indicates that the ambiguity stems from omitted content, where essential information is left out, leading to uncertainty about the intended meaning.
- *Colloquialism* refers to ambiguity arising from using informal or imprecise expressions that may not be universally understood or may vary in interpretation across different contexts.
- *Subjectivity* indicates ambiguities due to subjective judgments, where personal opinions or individual perspectives cause unclear or varied interpretations.
- *Relativity* indicates that ambiguity comes from implied comparisons, where the meaning depends on an unstated reference point or context, making the instruction open to multiple interpretations.
- *Other* encompasses all other types of ambiguity that do not fit neatly into the previously mentioned categories, covering a broad range of miscellaneous sources of confusion.

We will release the data we set up under the MIT license.

## C DEEPER ANALYSIS ON DATASET

To analyze the difficulty of the ambiguous dataset, we further conduct human experiments.

**Our dataset is relatively easy for humans but difficult for AI models.** We conduct a human experiment by inviting 10 volunteers to answer questions of our ambiguous dataset. The results are shown in the Table B.

Table B: **Human results on our dataset.**

| HUMAN | GPT-4O | LLAVA |
|-------|--------|-------|
| 82.17 | 14.10 | 8.58 |

Table C: **Average number of Q&A turns on different datasets.**

| METHOD | AMBIGUOUS | GENERAL |
|--------|-----------|---------|
| (A) HUMAN-GPT | 7.1 | 3.4 |
| (B) HUMAN-HUMAN | 4.8 | 2.2 |

Table D: **The improvements on different types of ambiguity.**

| METHOD | ELLIPSIS | | COLLOQUIALISM | | SUBJECTIVITY | | RELATIVITY | | OTHER | |
|--------|----------|------|---------------|------|--------------|------|------------|------|-------|------|
| | GIOU | CIOU | GIOU | CIOU | GIOU | CIOU | GIOU | CIOU | GIOU | CIOU |
| LISA | 0.0424 | 0.0460 | 0.0107 | 0.0124 | 0.0025 | 0.0081 | 0.0113 | 0.0119 | 0.0490 | 0.0633 |
| VISUAL-O1 | **0.1742** | **0.1704** | **0.1662** | **0.1375** | **0.0826** | **0.2637** | **0.1444** | **0.1207** | **0.0763** | **0.0490** |

We find that humans can correctly answer more than $80\%$ of the questions, indicating that the ambiguous dataset does not pose major challenges for humans. This is because humans have strong disambiguation abilities, enabling them to accurately understand the meaning of ambiguous instructions by combining them with images, a capability that AI models lack.

**The difficulty of our dataset lies in the image-text ambiguity.** We design another human experiment where volunteers can only read original instructions but are not allowed to see images. In each turn, volunteers are allowed to ask one question and receive one answer from either (a) GPT-4O, which can see the images but not the instructions, or (b) another volunteer who can also see the images but not the instructions. Each sample will go through up to 10 turns of Q&A or until the correct answer is obtained. Volunteers cannot ask questions that have the same meaning as the original instructions. We conduct the experiment on both the general dataset and the ambiguous dataset and calculate the average number of Q&A turns for samples that successfully obtained the correct answer.

As shown in Table C, we find that on our ambiguous dataset, the turns of Q&A greatly improved. This is because, for samples with ambiguity, volunteers have difficulty determining what exactly to ask, leading to numerous tentative questions before finally pinpointing the intended question and obtaining the correct answer. In contrast, on the general dataset, volunteers can often quickly identify what needs to be asked, resulting in much lower turns of Q&A. This also demonstrates that text-image ambiguity in our dataset increases the difficulty of reasoning.

## D    DEEPER ANALYSIS ON IMPROVEMENT

To deeply analyze how VISUAL-O1 mitigates ambiguity issues, we separately calculate the original scores and the scores after using VISUAL-O1 for each ambiguity category in RIS, as shown in Table D. We find that, in the original case, LISA performs relatively best in *ellipsis*, because *ellipsis* is the relatively simplest category, while the scores for the other categories are almost all around 1. After using Visual-O1, we observe notable improvements in scores across various categories, particularly in *colloquialism* and *relativity*. This is because these two can be more easily converted into clear instructions through visual context. The improvements in other more challenging categories also signify the effectiveness of VISUAL-O1.

Table E: **General instruction results on different intelligence level.**

| MODEL | VISUAL-O1 | GIoU | CIoU |
|---|---|---|---|
| LISA (LAI ET AL., 2023) | | 0.4654 | 0.4721 |
| VISUAL-O1 (LISA) | EMPIRICAL | **0.4738** | **0.4980** |
| VISUAL-O1$^+$ (LISA) | INSTANTIAL | **0.4985** | **0.5188** |
| SoM (YANG ET AL., 2023) | | 0.3507 | 0.4154 |
| VISUAL-O1$^-$ (SoM) | EMPIRICAL | **0.3772** | **0.4336** |
| VISUAL-O1 (SoM) | INSTANTIAL | **0.3686** | **0.4508** |

Table F: **Generalization studies on VLN.**

| MODEL | SR | SPL | NAVI ERROR |
|---|---|---|---|
| VLN-SIG (LI & BANSAL, 2023) | 4.72 | 4.59 | 7.95 |
| VISUAL-O1 (VLN-SIG) | **26.42** | **22.66** | **5.99** |

# E    FURTHER GENERALIZABILITY VERIFICATION ON INTELLIGENCE

We not only compare the results on ambiguous instructions but also extend the comparison to general instructions to better confirm the generalization capability of VISUAL-O1. In Table E, we can also see that all variants of VISUAL-O1 maintain and even improve performance on general instructions. This indicates that the design of VISUAL-O1 is highly flexible and can be broadly applied to various intelligence levels of models, thereby improving the whole system's reasoning efficiency.

# F    FURTHER GENERALIZABILITY VERIFICATION ON EXTRA TASKS

VISUAL-O1 demonstrates remarkable generalizability across various tasks. To further validate this, we apply VISUAL-O1 to two complex multi-modal tasks: image synthesis in the visual synthesis field and vision-and-language navigation (VLN) in the robotics field. Since models for these types of tasks do not have direct language output capabilities, we use GPT-4O for parts requiring intermediate language output. For VLN, we use the valid unseen split of the ROOM-TO-ROOM dataset (Anderson et al., 2018). As detailed in Table F, our observations reveal substantial performance enhancements across all tasks, underscoring the versatility and robustness of VISUAL-O1. We also provide examples in Figure A. For image synthesis, we choose the state-of-the-art model DALL-E 3. As shown in Figure B, we notice that even the most advanced models often misunderstand ambiguous instructions in human interactions, whereas VISUAL-O1 substantially alleviates this issue. This broad applicability paves the way for extending our method to a wider array of applications, showcasing its potential to improve performance in diverse contexts.

# G    EXTRA EXPLORATION OF THE MODEL

## G.1    EXPLICIT CLEAR INSTRUCTION

We slightly adjust the prompt to allow the reasoning process of instantial experience to generate clear instructions explicitly. We find that implementing this is very simple as it only needs to add a sentence to $p_{\text{syn}}$: `Explicitly generate clear instructions within <clear instruction> and </clear instruction> tags before answering.`

As shown in the Table G, we can see that explicitly generating clear instructions can further enhance the performance. We explain this as explicit clear instruction will help the model to review and better understand the instruction.

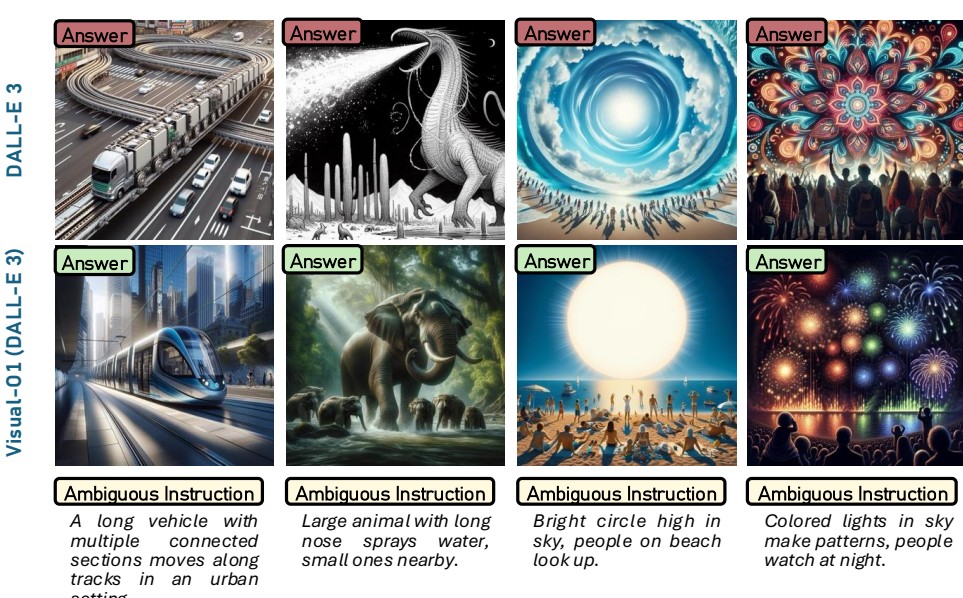

| **Image** | **VLN-SIG** | **Visual-O1 (VLN-SIG)** |
|---|---|---|

**Answer**

Success: no
Nav Error: 4.1363
SPL: 0.00

**Answer**

Success: yes
Nav Error: 1.7766
SPL: 1.00

Clear Instruction

*Move forward towards the open door that you see straight ahead. As you pass through the door, you will see a staircase to your right. Ascend the stairs and proceed until you reach the area set up for dining. Stop there.*

Ambiguous Instruction — *Please guide me up the stairs and stop once we reach where we eat.*

**Answer**

Success: no
Nav Error: 4.6138
SPL: 0.00

**Answer**

Success: yes
Nav Error: 0.0000
SPL: 0.99

Clear Instruction

*Move forward, heading towards the door past the foot of the bed. Once you reach the door, proceed through it to enter the hallway. Look for the nearest place to wash hands, which could be a bathroom or a sink, and go there to wash your hands. If the exact location to wash hands is not immediately visible, continue in the general direction along the hallway and turn right at the next opportunity to look for a bathroom or sink.*

Ambiguous Instruction — *Guide me past the bed to the hall, take me to wash hands.*

Figure A: **Cases of Visual-O1 on VLN.**

**DALL-E 3**

**Answer** | **Answer** | **Answer** | **Answer**

**Visual-O1 (DALL-E 3)**

**Answer** | **Answer** | **Answer** | **Answer**

Ambiguous Instruction | Ambiguous Instruction | Ambiguous Instruction | Ambiguous Instruction

*A long vehicle with multiple connected sections moves along tracks in an urban setting.* | *Large animal with long nose sprays water, small ones nearby.* | *Bright circle high in sky, people on beach look up.* | *Colored lights in sky make patterns, people watch at night.*

Figure B: **Cases of Visual-O1 on Image Synthesis.**

Table G: **Instantial experience with explicit clear instruction.**

| VISUAL-O1 | GIoU | CIoU |
|---|---|---|
| EMPIRICAL | 0.1143 | 0.1530 |
| INSTANTIAL | 0.1304 | 0.1756 |
| INSTANTIAL W/ EXPLICIT CLEAR INSTRUCTION | **0.1316** | **0.1864** |

Table H: **Comparisons of joint instruction.**

| VISUAL-O1 | GIoU | CIoU |
|---|---|---|
| W/O JOINT INSTRUCTION | 0.1131 | 0.1215 |
| W/ JOINT INSTRUCTION | **0.1245** | **0.1463** |

### G.2 JOINT INSTRUCTION

We expand the functionality of response synthesis: instead of directly integrating the clear instruction with the original instruction, we designed a simple prompt to use a general-intelligent model to rewrite the clear instruction into a joint instruction guided by the purpose of the original instruction, ensuring keeping the original purpose while integrating as much information from the clear instruction as possible.

As shown in Table H, we are surprised to find that the accuracy further increases. We explain this as the extra step can keep the original purpose by fixing some misalignment between clear and original instructions. This also aligns with the findings in our paper that the response synthesis is important for VISUAL-O1.

## H EXTRA EXPLORATION OF REASONING PROCESS

### H.1 COMPARISONS OF EXPERIENCE PROMPT

To better demonstrate VISUAL-O1's reasoning process and reveal the specific content of VISUAL-O1's reasoning, we show the detailed instantaneous and empirical experiences after each reasoning step of VISUAL-O1. The upper part is the instantaneous experience for the highly intelligent model, while the lower part is the empirical experience for the generally intelligent model. The sentences in gray are given by reflection.

---

**Experience Prompt of $\mathcal{A}_{\text{ins}}$ (Top) and $\mathcal{A}_{\text{emp}}$ (Bottom)**

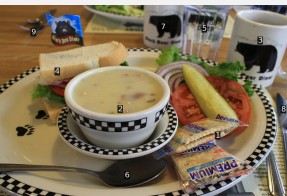

between two bears and not plate

**Budget 7** Identify the objects marked as bears in the image.

**Budget 6**
The bears in the image are part of the design on the mugs, which are marked with [7] and [3].

**Budget 5**
Locate objects that are not plates.  In this image, the plate is a prominent object, but we need to identify objects that

---

```
are not plates.
```

**Budget 4**
```
Exclude the objects that are on the plate, such as the
sandwich marked with [4], the soup bowl marked with [2], and
the pickle marked with [8].
```

**Budget 3**
```
Identify any objects that are between the two bears but are
not on the plate.
```

**Budget 2**
```
It seems that the description provided was to find an object
that is situated between the two bears but is not a plate
itself.  The steps taken so far have led to the exclusion of
items on the plate and the identification of the bears on the
mugs.
<reward> 0.8 </reward>
```

**Budget 1**
```
Looking at the objects, the mugs marked with [7] and [3] are
situated on either side of the image.  The only object that
is directly between them and not on the plate seems to be the
salt or pepper shaker marked with [5].
```

**Budget 0**
```
The object that fits the description "between two bears and
not plate" is the salt or pepper shaker marked with [5].
The process involved a systematic exclusion of objects on
the plate and identifying the bears on the mugs.  The final
object that fit the description was found to be between the
bears (mugs) and not on the plate.  The reasoning was sound,
and followed the instructions accurately.
<reward> 1.0 </reward>
```

**Budget 3** Repeat the question.

**Budget 2** Please review the image provided and use its
details to rephrase the ambiguous and blurry question into
a clear and precise one that can be effectively answered.
Ensure that your revised question is directly related to the
content of the image.

**Budget 1** Please ensure that your revised question is
specific and directly references observable elements in
the image.  Your question should guide the respondent in
addressing particular details or aspects present in the image
clearly despite any blurriness.  Avoid general or vague terms
and aim for specificity that will elicit a precise answer.

**Budget 0** When rephrasing the question, focus on the
discernible elements in the image, such as text, icons,
or specific features visible on the computer monitor.
Your question should ask for details about these specific
elements, avoiding any reference to the clarity of the
picture or the physical location, as these are not relevant

> to the content displayed on the screen.  Aim to formulate
> a question that inquires about the information or processes
> shown in the image, which can be answered with the visible
> data.

As the reasoning progresses, in the instantial experience, VISUAL-O1 begins to perform increasingly in-depth reasoning.  Meanwhile, in the empirical experience, VISUAL-O1 starts to generate more specific requirements, such as generating specific locations and elements, which are essential for accurately interpreting and generating unambiguous instructions. This also proves the effectiveness of VISUAL-O1's optimization.

## H.2   COMPARISONS OF REASONING TRAJECTORY

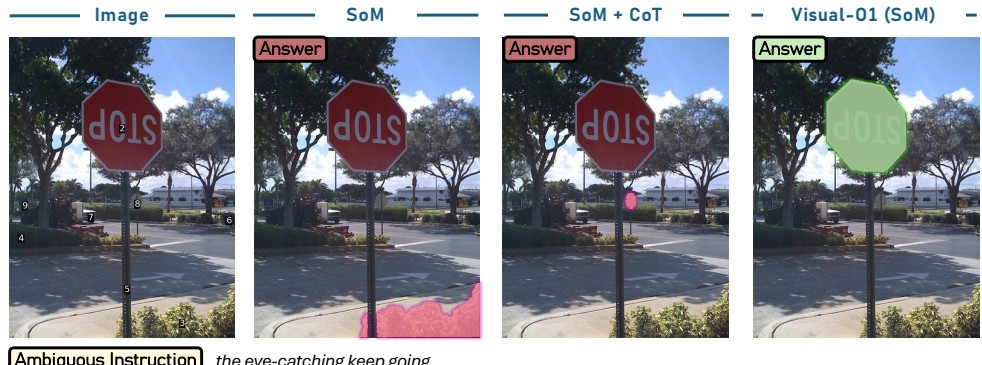

Figure C: **Common case of VISUAL-O1 with trajectory.**

---

**Trajectory of SOM + COT (Top) and VISUAL-O1 (SOM) (Bottom)**

```
The phrase "the eye-catching keep going" suggests finding
something that stands out and implies continuity or
progression.  In the context of this image with numbered
elements and the instruction, let's analyze the image step
by step:

The most eye-catching element in the image is the stop sign,
which is labeled with the number [2].
The phrase "keep going" implies progression or continuity.
In a sequence, we often continue to the next number.
Considering the instruction, we look at the number [8], which
is on the pole of the stop sign.

Therefore, the id of "the eye-catching keep going" is [8].
```
- - - - - - - - - - - - - - - - - - - - - - - - - - - - - - - - - - - - - - -
```
<count> 5 </count>
<step> Examine the image for elements that might be described
as "eye-catching keep going." </step>

<count> 4 </count>
<step> Identify the most prominent and eye-catching object
in the image.  The upside-down stop sign (with ID [2]) is the
most eye-catching object in the image.  </step>
<reflection> The steps taken so far correctly identify the
most prominent and eye-catching object in the image.  Since
```

Table I: **Ablation studies.**

| MODEL | ACC | BLEU-1 |
|---|---|---|
| VISUAL-O1 (LLAVA) | **57.58** | **0.7295** |
| W/O MULTI-EXAMPLES | 55.75 | 0.7092 |
| W/O MULTI-MODALITIES | 55.90 | 0.7143 |

```
the stop sign is upside down, it draws attention and might
be interpreted as "keep going" because it is not in its usual
orientation.  Therefore, it is reasonable to assume that the
stop sign is the target object.  </reflection>
<reward> 1.0 </reward>

<count> 3 </count>
<step> Identify the number associated with the most prominent
and eye-catching object that contradicts its usual meaning,
suggesting "keep going" instead of stopping.  </step>

<count> 2 </count>
<step> Confirm that the upside-down stop sign is the focus of
the instruction, as it is the prominent object that suggests
an ironic "keep going" message.  </step>
<reflection> The steps and process effectively identified
the upside-down stop sign as the most eye-catching object due
to its unusual orientation.  This sign draws attention and
suggests an ironic "keep going" message.  The identification
of the stop sign's ID as [2] was made accurately, and
the reasoning aligns well with the given instruction.
</reflection>
<reward> 0.9 </reward>

<count> 1 </count>
<step> Verify the ID on the upside-down stop sign, which is
[2].  </step>

<count> 0 </count>
<answer> [2] </answer>
<reflection> The steps taken effectively identified the most
eye-catching object that suggests "keep going" due to its
unusual orientation.  The interpretation aligns well with the
instruction, and the ID [2] has been correctly identified as
the target.  </reflection>
<reward> 1.0 </reward>
```

# I ADDITIONAL ABLATION STUDIES

## I.1 COMPONENTS

We further verify the effectiveness of VISUAL-O1's one-time optimization design details in Table I. We find that without multiple examples, *i.e.*, only allowing optimization for the same sample leads to overfitting and ineffective optimization. Without multiple modalities, *i.e.*, lacking visual contexts, VISUAL-O1's optimization capability also declines, as visual contexts help the optimizer determine the current state of optimization. By avoiding all these issues, it demonstrates excellent capabilities.

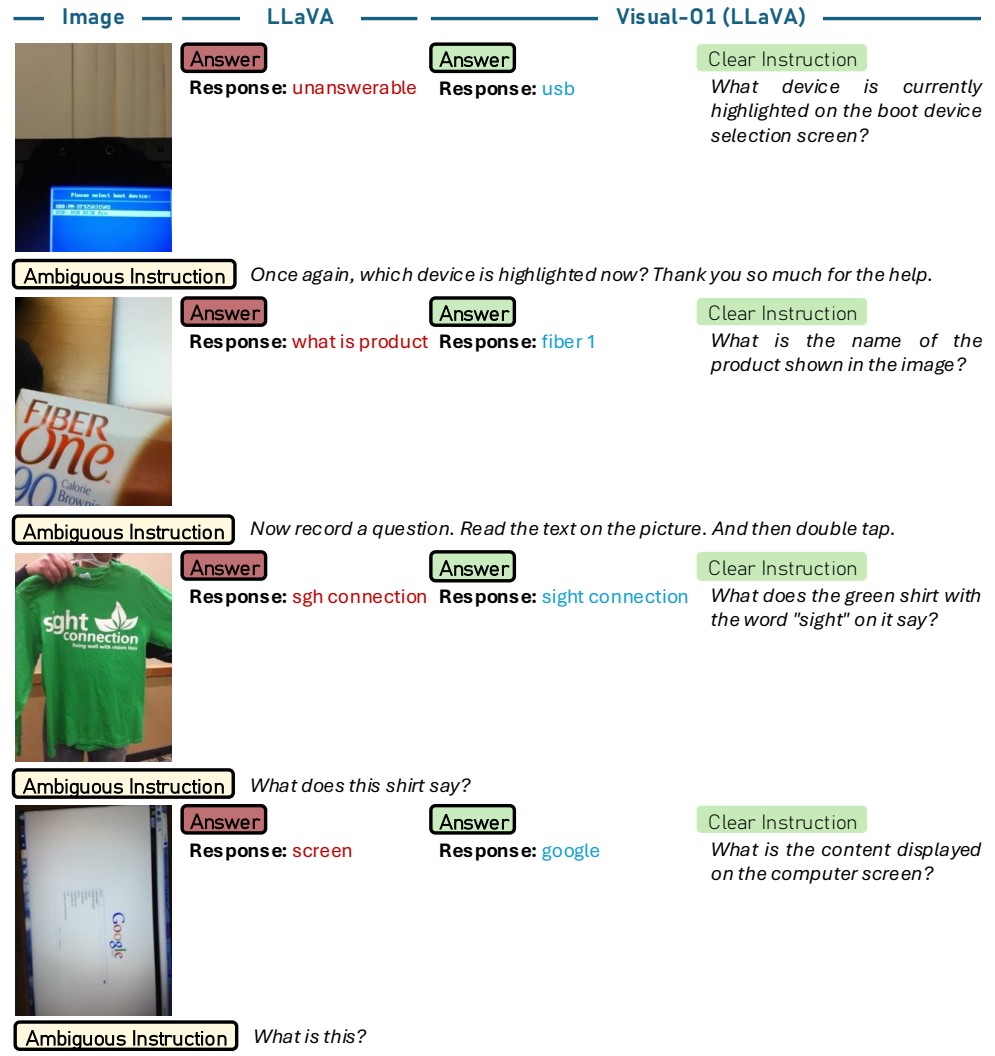

Figure D: **Case studies on VQA.**

Table J: **Results of different budgets with summarization.**

| VISUAL-O1 | AMBIGUOUS | | | | GENERAL | | | |
|---|---|---|---|---|---|---|---|---|
| | AVG | MAX | MIN | VAR | AVG | MAX | MIN | VAR |
| W/O BUDGET | 8.95 | 10.14 | 7.89 | 0.85 | 54.02 | 54.90 | 53.30 | 0.35 |
| W/ HUMAN EXP | 16.88 | 18.43 | 14.29 | 1.61 | 54.55 | 56.20 | 53.15 | 1.02 |
| W/ 1 BUDGET | 17.83 | 19.73 | 16.76 | 0.96 | 55.38 | 56.22 | 54.10 | 0.45 |
| W/ 2 BUDGET | 19.92 | 21.89 | 18.77 | 0.95 | 56.31 | 56.92 | 54.16 | 0.58 |
| W/ 3 BUDGET | 22.13 | 24.52 | 20.91 | 0.88 | 57.58 | 58.56 | 56.76 | 0.35 |
| W/ 4 BUDGET + SUMM | 22.90 | 25.13 | 21.43 | 0.88 | 57.60 | 58.57 | 56.95 | 0.25 |
| W/ 5 BUDGET + SUMM | 23.00 | 25.28 | 21.78 | 0.69 | 57.79 | 58.62 | 56.84 | 0.28 |

## I.2 REASONING BUDGET

The performance decrease observed when the budget $N > 3$ is due to the length of empirical experience prompt becoming longer, making it difficult for the current general-intelligent model to perform image-text understanding effectively. However, this is not an unsolvable problem. By

Table K: **Computational overhead of Visual-O1.**

| METHOD | STAGE | TIME | VRAM |
|---|---|---|---|
| LLAVA | LORA FINE-TUNING (HU ET AL., 2021) | 25.4439MIN | 88797MB |
| VISUAL-O1 (LLAVA) | ONE-TIME OPTIMIZATION | 1.5260MIN | 16148MB |
| LLAVA | INFERENCE | 0.5103s | 16102MB |
| VISUAL-O1 (LLAVA) | INFERENCE | 0.6547s | 16370MB |
| GPT-4O | INFERENCE | 4.9403s | - |
| VISUAL-O1 (GPT-4O) | INFERENCE | 6.7614s | - |

adding an extra step where the model summarizes and simplifies long prompts, the prompt length can be shortened.

As shown in Table J, we find that the accuracy further improves when the budget $N > 3$. This means that if the model understands long texts well, our method can achieve increasingly higher results as the budget increases. If the model can only understand shorter texts, a budget of 3 is sufficient to achieve good results.

## J ADDITIONAL COMPUTATION STUDIES

As shown in Table K, we analyze the computational overhead of VISUAL-O1. We find that VISUAL-O1's overhead in both the optimization and disambiguation phases is low. Specifically, we also compare the time it takes for the vanilla LLAVA and GPT-4O to complete a VQA task. The computational overhead is comparable to LLAVA, further proving VISUAL-O1's capability.

## K ADDITIONAL CASE STUDIES

### K.1 VQA CASE

Similar to RIS, we also visualize examples on VQA. As shown in Figure D, VISUAL-O1 enhances the clarity of ambiguous instructions, helping the task model produce more accurate and natural results.

### K.2 FAILED CASE

To figure out how VISUAL-O1 fails in some cases, we conduct the analysis on failed cases. We find that these errors can generally be divided into two categories: (1) errors caused by non-reasoning factors such as calculation, color recognition, and orientation identification errors, and (2) errors caused by excessive reasoning.

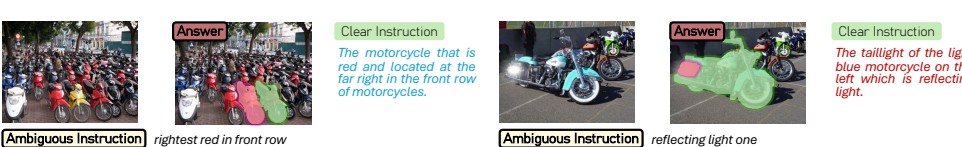

Figure E: **Failed cases of VISUAL-O1.**

The first type of error is shown in the left part of Figure E, we find that although VISUAL-O1 helps the model perform reasoning analysis well and generates clear instructions, the model is still limited by its segmentation capabilities, resulting in incorrect outcomes. However, we believe that as the model's understanding ability improves, such errors will gradually decrease.

For the second type of error, we find that sometimes VISUAL-O1 leads to over-reasoning. As shown in the right part of Figure E, we find that VISUAL-O1 over-interprets the meaning of "reflecting

`light one`", thinking it refers to a small reflective area on the motorcycle rather than the whole motorcycle. However, the instruction refers to the entire motorcycle. Despite the presence of excessive reasoning in some samples, the overall better performance on the general dataset means that, in most cases, VISUAL-O1 can effectively reason out the correct meaning of the instructions.

## K.3 CHALLENGING CASE

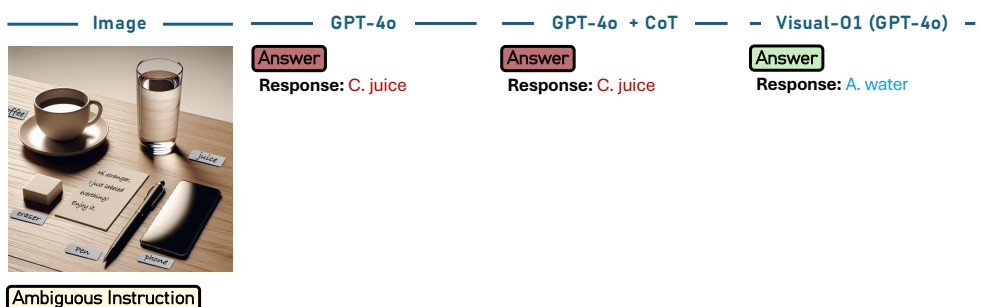

Figure F: **Challenging case of VISUAL-O1 with trajectory.**

Inspired by the theory of mind, we design a very challenging case in Figure F. We compare GPT-4O, GPT-4O + COT, and our VISUAL-O1. We find that GPT-4O and GPT-4O + COT respond to the wrong answer: `C. juice`, indicating their weak understanding of ambiguous multimodal scenarios.

By reviewing the trajectory of GPT-4O + COT, we note that GPT-4O + COT recognized that Lisa incorrectly labeled the water cup but mistakenly believed that Lisa thought the cup contained juice because she placed the label. Given that the cup is transparent in the image, we know this is impossible, meaning that although COT enhances reasoning abilities, its effect is limited in ambiguous multimodal scenarios.

As shown in the trajectory of VISUAL-O1, it correctly understood that the question was actually asking what Lisa thought the liquid in the glass cup was. Then, noticing the amusing note left by Lisa on the table, it inferred that Lisa intentionally mislabeled the items. Thus, it realized that Lisa knew the cup contained water and gave the correct answer, `A. water`, correctly interpreting the ambiguous command in the multimodal context.

---

**Trajectory of GPT-4O + COT (Top) and VISUAL-O1 (GPT-4O) (Bottom)**

```
Let's analyze the situation step by step:

1.  Items and Labels:
- Lisa poured a glass of water and labeled it as "juice".
- Lisa poured a porcelain cup of coffee and labeled it
correctly as "coffee".
- Lisa prepared a pen and labeled it correctly as "pen".
- Lisa prepared a mobile phone and labeled it correctly as
"phone".
- Lisa prepared an eraser and labeled it correctly as
"eraser".

2.  Correctness of Labels:
- The "coffee" label is correct.
- The "pen" label is correct.
- The "phone" label is correct.
```

```
- The "eraser" label is correct.
- The "juice" label is incorrect because it is actually a
glass of water.

Given the labels, the only thing that is substantially
incorrect is labeling the glass of water as "juice". Lisa
labeled it as "juice" as she thinks it is "juice".

Thus, Lisa thinks it substantially be:  C. juice
- - - - - - - - - - - - - - - - - - - - - - - - - - - - - - - - - - - - - - -
<count>6</count>
<step>Observe the given image carefully.  Notice that Lisa
has labeled each item on the table.  The labels are "coffee"
for the cup of coffee, "eraser" for the eraser, "pen" for the
pen, "phone" for the mobile phone, and "juice" for the glass
of water.</step>
<reflection>Start by identifying each item on the table and
their corresponding labels.</reflection>
<reward>0.8</reward>

<count>5</count>
<step>Identify the discrepancy between the items and their
labels.  The glass of water is labeled as "juice," which is
incorrect.</step>
<reflection>Recognize the mismatched label to understand the
source of confusion or humor in the situation.</reflection>
<reward>0.9</reward>

<count>4</count>
<step>Consider the context of the amusing note left by Lisa,
which states, "Hi stranger, I just labeled everything!  Enjoy
it." This implies that the labeling is meant to be humorous
or misleading.</step>

<count>3</count>
<step>Determine the logical conclusion from the context and
the incorrect label.  The strange thing Lisa might think it
substantially be is the mislabeled glass of water.</step>
<reflection>Connect the incorrect labeling to the
context to identify what Lisa might find strange or
amusing.</reflection>
<reward>1.0</reward>

<count>2</count>
<step>Match the options provided (A. water, B. coffee,
C. juice, D. eraser, E. phone) with the conclusion.  The
glass of water being mislabeled as "juice" aligns with the
question's context.</step>

<count>1</count>
<step>Choose the correct option based on the analysis.  The
glass of water is the item Lisa thinks is strange due to its
incorrect label.</step>

<count>0</count>
<answer>A. water</answer>
```

```
<reflection>The solution process involved carefully observing
the image, identifying the incorrect label, and understanding
the context of the note.  The logical conclusion was
drawn based on the humor implied by the note and the
mismatch between the item and its label.  The final answer
correctly identifies the glass of water as the strange
item.</reflection>
<reward>0.9</reward>
```

## L  BROADER IMPACT

With the development and application of AI, language instructions for interacting with AI are being applied in an increasing number of scenarios. In reality, humans often issue vague instructions for communication, and due to their natural visual ability, language and visual information complement each other, which further intensifies the ambiguity of instruction information. In this paper, we reveal the phenomenon of ambiguous instructions. Our proposed VISUAL-O1 substantially alleviates the pressure of ambiguous instructions on AI models, helps AI understand more natural instructions, broadens the application range of AI, and serves more non-professional people.

## M  LIMITATIONS

Despite VISUAL-O1 demonstrating strong reasoning capabilities and allowing general-intelligent models like LLAVA to perform inference without relying on large-scale high-intelligent models such as GPT-4O, it still requires the use of high-intelligent models during the one-time optimization phase. In the future, we will explore methods that do not rely on high-intelligent models even during optimization, in order to better extend our approach to scenarios with more limited resources.

## N  REPRODUCIBILITY STATEMENT

We place a high emphasis on the reproducibility of our work. To facilitate this, we have provided a comprehensive set of implementation details and prompts in the appendix. Additionally, to further enhance the reproducibility of our results, we will release the source code and data.

