# OpenReview forum: "Visual-O1: Understanding Ambiguous Instructions via Multi-modal Multi-turn Chain-of-thoughts Reasoning"
_ICLR.cc/2025/Conference — ICLR 2025 Poster_

### Official Review · Reviewer_FebY · 2024-11-03

**Soundness:** 3
**Presentation:** 3
**Contribution:** 3
**Rating:** 8
**Confidence:** 4

**Summary:**

The paper proposes a method for visual o-1 that is similar to language o1 that increases the computation budget on test-time. The work is proposed to highlight the potential of artificial intelligence to work like humans in the real world, especially when the instruction is ambiguous for the instructional model. This is achieved by multi-turn reasoning with chain of thought.

They have separated the multi-round reasoning process for high-intelligence model and general intelligent model, where it tends to have more round interaction when the model has a higher intelligence level.

They have evaluated their result on visual-QA, and RIS in comparison with the state-of-the-art model and shows reasonable improved on both segemtantion and visual-QA.

**Strengths:**

1. The paper's proposed direction of Visual-O1 towards instructional QA is an interesting direction.
2. By integrating the chain of thought into the pipeline, the model can improve the segmentation result of referring image segmentation, and the visual QA result. The performance gain on visual QA is larger than RIS, and the performance gain on ambiguous cases is higher than general cases, which indicates that the multi-round and multimodal inference needs more on the harder case.
3. The additional overhead of the visual chain of thought is limited for both training and evaluation stages.

**Weaknesses:**

1. To be completely honest, although the performance gain shows statistically significant improvement, however, the improvement is marginal, which indicates that the chain-of-thought works.
2. The author lacks examples when we actually need visual CoT (As the cipher example in o1).
3. As shown in Table.6, I actually think the budget curve is confusing, the performance fluctuates for Acc and BLEU, and saturate w/ 3 budget, this indicates that the designed algorithm is not sophisticated enough for scaling up.
4. They propose the empirical solution for less-intelligent model.

**Questions:**

1. Can you write the exact GPU settings for Table M?
2. How does the author interpret that the performance saturates quickly with the budget?

---

> ### Author Response · Authors · 2024-11-20
> **Authors Response 1/2**
>
> Thank you for your valuable advice and comments!
>
> ## Q1: The Improvement
> This is because instructions in the general dataset mostly do not contain ambiguity. So, the improvement of Visual-O1 on the general dataset is stable but not as obvious as an ambiguous dataset. However, on the ambiguous dataset, we achieved improvements over the baseline model with LISA: $+344\%$, SoM: $+48\%$, LLaVA: $+149\%$, and GPT-4o: $+57\%$.
>
> Additionally, we find that although traditional CoT can bring some improvements to the ambiguous dataset, it causes a observable performance drop on the general dataset due to the lack of explicit multi-turn visual reasoning and reflection processes. Our Visual-O1 not only brings the largest improvement on the ambiguous dataset but also provides stable improvement on the general dataset, showing the practicality of our method.
>
> ## Q2: Challenging Case like Cipher
>
> Thanks for your great idea. We provide a challenging case like o1's cipher to demonstrate the effect of Visual-O1:
>
> Inspired by the theory of mind, we design a very challenging case. Refer to the Figure F of the appendix, there is a series of objects on the table: a phone, a water-filled cup, a coffee-filled cup, a pen, and a telephone. The water cup is transparent, clearly showing the transparent liquid inside. All objects have labels with their names, but the label next to the water cup shows "juice". The model will be asked the question: "*Lisa poured a glass of water, a porcelain cup of coffee, and prepared a pen, a mobile phone, and an eraser for her friend's visit. Then, Lisa labeled all the items and left an amusing note. But here's something strange for her friend. What does Lisa think it substantially be? A. water   B. coffee   C. juice   D. eraser   E. phone*".
>
> We compare GPT-4o, GPT-4o + CoT, and our Visual-O1. We find that GPT-4o and GPT-4o + CoT respond to the wrong answer: C. juice, indicating their weak understanding of ambiguous multimodal scenarios.
>
> By reviewing the trajectory of GPT-4o + CoT, we note that GPT-4o + CoT recognized that Lisa incorrectly labeled the water cup but mistakenly believed that Lisa thought the cup contained juice because she placed the label. Given that the cup is transparent in the image, we know this is impossible, meaning that although CoT enhances reasoning abilities, its effect is limited in ambiguous multimodal scenarios.
>
> As shown in the trajectory of Visual-O1, it correctly understood that the question was actually asking what Lisa thought the liquid in the glass cup was. Then, noticing the amusing note left by Lisa on the table, it inferred that Lisa intentionally mislabeled the items. Thus, it realized that Lisa knew the cup contained water and gave the correct answer, A. water, correctly interpreting the ambiguous command in the multimodal context.
>
> ## Q3: Budget Curve
>
> In the previous version of Table 6 of the paper, we reported an extreme sample on the general dataset to show that even if there are slight fluctuations in the reasoning process, we can still return to an effective reasoning process. Additionally, since the general dataset contains mostly unambiguous instructions, the performance improvement is not as large as that of the ambiguous dataset.
>
> To further demonstrate the stability of our method, we extend the experiment to both ambiguous and general datasets, conducting $10$ experiments with different random seeds. We fixed the random seeds to $40$, $41$, $42$, $43$, $44$, $45$, $46$, $47$, $48$, and $49$. As shown in the tables below, we report the average accuracy of all experiments, as well as their maximum, minimum, and variance.
>
> | Budget | Mean   | Max   | Min   | Div   |
> | ----   | ------ |------ |------ |------ |
> | 1      | 55.38  |	56.22 |	54.10 |	0.45  |
> | 2      | 56.31  |	56.92 |	54.16 |	0.58  |
> | 3      | 57.58  |	58.56 |	56.76 |	0.31  |
> | 4      | 55.42  |	56.34 |	54.36 |	0.34  |
> | 5      | 56.17  |	57.44 |	55.14 |	0.46  |
>
> | Budget | Mean   | Max   | Min   | Div   |
> | ----   | ------ |------ |------ |------ |
> | 1      | 17.83  |	19.73 |	16.76 |	0.96  |
> | 2      | 19.92  |	21.89 |	18.77 |	0.95  |
> | 3      | 22.13  |	24.52 |	20.91 |	0.78  |
> | 4      | 18.82  |	20.54 |	17.30 |	0.84  |
> | 5      | 19.85  |	21.35 |	17.94 |	0.95  |
>
> This demonstrates that our method exhibits good stability on the ambiguous dataset and maintains a positive effect on the general dataset.
>
> For further performance analysis with a budget $N > 3$, please refer to Q6.
>
> (Continued in Authors Response 2/2)

---

> ### Author Response · Authors · 2024-11-20
> **Authors Response 2/2**
>
> (From the Authors Response 1/2)
>
> ## Q4: Empirical Solution for General-intelligent Model
>
> The reason we design the empirical experience for the general-intelligent model is that the general-intelligent model has limited text generation and reasoning capabilities, making it difficult to perform instantial experience that requires long text reasoning like a high-intelligent model such as GPT-4o.
>
> However, as shown in Table 3 of the paper, we find that if the general-intelligent model uses GPT-4o to help it perform long text reasoning, it also has the potential to use instantial experience and show promising results. This further proves our Visual-O1's flexibility for models with different levels of intelligence.
>
> ## Q5: GPU Setting
> We use the Nvidia RTX 3090 in all experiments for fair comparisons.
>
> ## Q6: Performance Saturates of the Budget
>
> The performance decrease observed when the budget $N > 3$ is due to the length of empirical experience prompt becoming longer, making it difficult for the current general-intelligent model to perform image-text understanding effectively. However, this is not an unsolvable problem. By adding an extra step where the model summarizes and simplifies long prompts, the prompt length can be shortened, and we find that the accuracy further improves when the budget $N > 3$:
>
> | Budget | Mean   | Max   | Min   | Div   |
> | ----   | ------ |------ |------ |------ |
> | 1      | 55.38  |	56.22 |	54.10 |	0.45  |
> | 2      | 56.31  |	56.92 |	54.16 |	0.58  |
> | 3      | 57.58  |	58.56 |	56.76 |	0.31  |
> | 4 + Summarization | 57.60  |	58.57 |	56.95 |	0.25  |
> | 5 + Summarization | 57.79  |	58.62 |	56.84 |	0.28  |
>
> | Budget | Mean   | Max   | Min   | Div   |
> | ----   | ------ |------ |------ |------ |
> | 1      | 17.83  |	19.73 |	16.76 |	0.96  |
> | 2      | 19.92  |	21.89 |	18.77 |	0.95  |
> | 3      | 22.13  |	24.52 |	20.91 |	0.78  |
> | 4 + Summarization | 22.90  |	25.13 |	21.43 |	0.88  |
> | 5 + Summarization | 23.00  |	25.28 |	21.78 |	0.69  |
>
> This means that if the model understands long texts well, our method can achieve increasingly higher results as the budget increases. If the model can only understand shorter texts, a budget of $3$ is sufficient to achieve good results.

---

> ### Author Response · Authors · 2024-11-23
> **We are very looking forward to discussing**
>
> Thank you again for your reviewing. Do you have any remaining concerns about our paper or response? We are very looking forward to discussing with you.

---

> ### Author Response · Authors · 2024-11-30
> **Are there any further concerns about our paper?**
>
> Thank you once again for your comments! Are there any remaining concerns about our paper? We would be very happy to address any questions or issues you may have.

---

> > ### Comment · Reviewer_FebY · 2024-12-02
> >
> > I appreciate the feedback from the authors, it solves all my concerns, especially for the challenging case and the budget curve. Although I think the claim of the title of Visual-O1 is broader than what this paper does, base on the solid experiment, I will recommend accepting this paper.

---

> > > ### Author Response · Authors · 2024-12-02
> > > **Thank You!**
> > >
> > > We greatly appreciate your thorough review and help. We will continue to polish our paper into a better shape!
> > >
> > > Thank you again!
> > >
> > > Authors

---

### Official Review · Reviewer_B9dk · 2024-11-03

**Soundness:** 3
**Presentation:** 2
**Contribution:** 3
**Rating:** 6
**Confidence:** 3

**Summary:**

This paper proposes Visual-O1, a multimodal multi-turn Chain-of-Thought reasoning framework to deal with ambiguous natural language instructions in two typical vision-language tasks: referring image segmentation and visual question answering. Visual-O1 builds
instance-specific experience for high-intelligence models or creates general experience for any ambiguous instructions for general-intelligence models. Experiments show that Visual-O1 improves the performance on both the ambiguous datasets and the general datasets.

**Strengths:**

1. This paper proposed a prompting method that seamlessly applies to both high-intelligence and general-intelligence models within the same framework. This adaptability indicates that Visual-O1 is not limited to a specific model type and can scale across different levels of model intelligence, which enhances its utility for a wider range of applications and users.
2. The authors presented extensive experimental results that include a variety of ablation studies and model comparisons. These results strengthen the credibility of their claims and provide thorough evidence of the framework's effectiveness. Notably, Visual-O1 not only improves performance on ambiguous instructions datasets but also enhances results on general datasets.

**Weaknesses:**

1. Although Visual-O1 shows clear improvements over baselines, the absolute success rates and IoU scores (on the ambiguity dataset) are still not entirely satisfactory. This gap suggests that there are underlying limitations to the model’s ability to handle certain types of ambiguous instructions. The paper could be strengthened if the authors provided a more detailed examination of failure cases (e.g., identifying specific patterns in instructions that remain challenging to disambiguate). Without this analysis, Visual-O1's readiness for real-world applications remains in question.
2. The paper tends to overemphasize its successes by frequently using terms like "significant," even when sometimes the improvements are marginal (e.g., success rate gains of less than 3%). This could be misleading, especially considering that the ambiguous instructions dataset is not large enough to robustly support such claims. Furthermore, the results do not report averages over different random seeds, which raises questions about the stability and generalizability of the improvements.
3. Minor issues or typos: The term $x_\text{rsn}$ should be $x_\text{rfl}$ in Eq. (6). The word "botlle" should be "bottle" in Figure 3.

**Questions:**

1. Could you provide some further information to explain the difficulty of the ambiguous dataset? For example, how humans (might) perform on this dataset?
2. How are the Chain-of-Though and FuDD baselines implemented?
3. An example of $\mathcal{A}_\text{ins}$ is provided in the appendix. If I understand the method correctly, it should be composed of the alternating appearance of reasoning and reflection. Are the sentences in gray given by reflection? If so, is there an example of the full trajectory?
4. Analysis of Visual-O1's failure modes will help a lot, as discussed in the weaknesses section.

---

> ### Author Response · Authors · 2024-11-20
> **Authors Response 1/2**
>
> Thanks for your invaluable advice!
>
> ## Q1: Analysis of Failure Cases
> We have now added an error analysis. We find that these errors can generally be divided into two categories:
> - Errors caused by non-reasoning factors such as calculation, color recognition, and orientation identification errors.
> - Errors caused by excessive reasoning.
>
> The first type of error is shown in the left part of Figure E of the appendix, we find that although Visual-O1 helps the model perform reasoning analysis well and generates clear instructions, the model is still limited by its segmentation capabilities, resulting in incorrect outcomes. However, we believe that as the model's understanding ability improves, such errors will gradually decrease.
>
> For the second type of error, we find that sometimes Visual-O1 leads to over-reasoning. As shown in the right part of Figure E  of the appendix, we find that Visual-O1 over-interprets the meaning of "reflecting light one," thinking it refers to a small reflective area on the car rather than the whole car. However, the instruction refers to the entire car. Despite the presence of excessive reasoning in some samples, the overall better performance on the general dataset means that, in most cases, Visual-O1 can effectively reason out the correct meaning of the instructions.
>
> ## Q2: The Improvement
> We conduct experiments on both ambiguous and general datasets. The improvement of Visual-O1 on the general dataset is stable but not as obvious as an ambiguous dataset. However, on the ambiguous dataset, we achieved improvements over the baseline model with LISA: $+344\%$, SoM: $+48\%$, LLaVA: $+149\%$, and GPT-4o: $+57\%$.
>
> ## Q3: Terms of Writing
> Thank you for your suggestions. We have revised our statements and will be more cautious about this in the future.
>
> ## Q4: Average Results over Different Random Seeds
> Thanks for your suggestions. We now report the average results of experiments on $10$ random seeds ($40$, $41$, $42$, $43$, $44$, $45$, $46$, $47$, $48$, $49$) and conducted statistical tests.
>
> | Setting           | Ambiguous-gIOU | Ambiguous-cIoU | General-gIOU | General-cIoU |
> | ----------------- | -------------- | -------------- | ------------ | ------------ |
> | Visual-O1 (LISA)   | 0.1131         | 0.1215         |  0.4738      | 0.4980       |
> | Visual-O1 (SoM)    | 0.1304         | 0.1756         |  0.3686      | 0.4508       |
>
> | Setting           | Ambiguous-Acc  | Ambiguous-BLEU-1 | General-Acc  | General-BLEU-1 |
> | ----------------- | -------------- | ---------------- | ------------ | -------------- |
> | Visual-O1 (LLaVA)  | 22.13          | 0.5095           | 57.58        | 0.7295         |
> | Visual-O1 (GPT-4o) | 22.78          | 0.6640           | 63.14        | 0.8482         |
>
> We find that the improvement is statistically significant with $p < 0.01$ under $t$-test, indicating that Visual-O1 has good stability and generalizability.
>
> ## Q5: Typos
> Thank you for your reminder. We have revised Eq. (6). In Figure 3 of the paper, we show the original instruction, which may contain some typos from the dataset, where "bottle" is misspelled as "botlle." To make this clearer, we have noted this in the caption.
>
> (Continued in Authors Response 2/2)

---

> ### Author Response · Authors · 2024-11-20
> **Authors Response 2/2**
>
> (From the Authors Response 1/2)
>
> ## Q6: Difficulty of the Ambiguous Dataset
> We further demonstrate the difficulty of our dataset from two perspectives:
>
> ### (1) Our dataset is relatively easy for humans but difficult for AI models.
> We conduct a human experiment by inviting $10$ volunteers to answer questions of our ambiguous dataset. The results are shown in the table below:
>
> | Setting   | Acc    |
> | --------- | ------ |
> | Human     | 82.17  |
> | GPT-4o    | 14.10  |
> | LLaVA     | 8.58   |
>
> We find that humans can correctly answer more than $80\%$ of the questions, indicating that the ambiguous dataset does not pose major challenges for humans. This is because humans have strong disambiguation abilities, enabling them to accurately understand the meaning of ambiguous instructions by combining them with images, a capability that AI models lack.
>
> ### (2) The difficulty of our dataset lies in the image-text ambiguity.
>
> We design another human experiment where volunteers can only read original instructions but are not allowed to see images. In each turn, volunteers are allowed to ask one question and receive one answer from either (a) GPT-4o, which can see the images but not the instructions, or (b) another volunteer who can also see the images but not the instructions. Each sample will go through up to $10$ turns of Q&A or until the correct answer is obtained. Volunteers cannot ask questions that have the same meaning as the original instructions. We conduct the experiment on both the general dataset and the ambiguous dataset and calculate the average number of Q&A turns for samples that successfully obtained the correct answer:
>
> | Setting         | General | Ambiguous |
> | --------------- | ------- | --------- |
> | (a) Human-GPT   | 3.4     | 7.1       |
> | (b) Human-Human | 2.2     | 4.8       |
>
> We find that on our ambiguous dataset, the turns of Q&A greatly improved. This is because, for samples with ambiguity, volunteers have difficulty determining what exactly to ask, leading to numerous tentative questions before finally pinpointing the intended question and obtaining the correct answer. In contrast, on the general dataset, volunteers can often quickly identify what needs to be asked, resulting in much lower turns of Q&A. This also demonstrates that text-image ambiguity in our dataset increases the difficulty of reasoning.
>
> ## Q7: Implementation of Baselines
> For FuDD, the model will find the five relevant objects of ambiguous instructions in the image. Then, the model will generate the key distinguishing information for each pair of objects. The model will use this disambiguating information appending to the question to obtain a better answer. We can find that FuDD focuses more on the ambiguity of similar language and visual content than explicitly considering the ambiguity that requires reasoning with visual context.
>
> For CoT, we use the classic prompt: "*Think it step by step.*" and our task-modified prompt: "*Analyze the question based on the image.*" There are slight differences in performance of two prompts and we report the higher scores in our experiments. We can find that CoT focuses more on reasoning at the language level but lacks the explicit process of reflection on multi-turn and multi-modal reasoning.
>
> ## Q8: Gray Sentence and Trajectory
>
> Yes, the sentences in gray are given by reflection. In practice, models sometimes do not engage in reflection at every step. We have observed that in some highly certain steps, the model tends to skip reflection, which also helps to speed up the reasoning process.
>
> We also provide two additional groups of raw trajectory: one for a normal case and one for a challenging case, compared with CoT without any highlighting, as shown in Figures C and F of the appendix.

---

> ### Author Response · Authors · 2024-11-23
> **Is there any remaining concerns about our paper?**
>
> Thanks again for your comments! Is there any remaining concerns about our paper? We are more than delighted to address any concerns/questions you may have.

---

> > ### Comment · Reviewer_B9dk · 2024-11-25
> >
> > Thank the authors for their extensive results and detailed responses, which enhance the clarity and strength of their work. Based on these improvements, I am increasing my rating.

---

> > > ### Author Response · Authors · 2024-11-25
> > > **Thank You!**
> > >
> > > We are very appreciated for your thorough review and helping the paper into a better shape!
> > >
> > > Thank you again!
> > >
> > > Authors

---

### Official Review · Reviewer_SvAq · 2024-11-04

**Soundness:** 2
**Presentation:** 3
**Contribution:** 3
**Rating:** 6
**Confidence:** 4

**Summary:**

This paper proposes Visual-O1, an approach that improves VLM's reasoning performance given ambiguous language instructions. For VLMs with stronger reasoning ability, the paper proposes an "instantial experience" approach where multi-turn multi-modal CoTs are generated to reason about the instruction, and the model subsequently directly outputs final answer conditioned on such reasonings. For VLMs with weaker reasoning ability, the paper proposes an "empirical experience" approach where after multi-modal CoTs are generated, explicit and clear instructions are generated before the model outputs the final answer. The authors demonstrate the effectiveness of their approach in referring image segmentation and visual question answering benchmarks.

**Strengths:**

- Context and instruction ambiguity is very prevalent in language and vision-language applications. Handling these cases is an important research problem. The proposed approach presents a promising way to improve the accuracy of VLM's "best guess reasoning" under ambiguous instructions.
- The paper is overall well-written, with clear methodology and extensive experiments.

**Weaknesses:**

- Table 6 indicates that the effectiveness of the proposed approach and the results of the paper may be vulnerable to noise, and the improvements could lack statistical significance. Specifically, as the budget limit increases from 1 to 5, the success rates are "55.87, 54.16, 57.38, 55.27, 55.56," showing no clear trend of performance improvement with respect to budget limit. For instance, the fact that "budget limit = 3" outperforms "budget limit = 2" by 3% and "budget limit = 4" by 2% could simply reflect evaluation noise.
- The generated "clear instruction" in Figure 3 (which correspond to authors' "empirical experience" approach) appears to directly answer the initial question, rather than serving as an actual "instruction" that enables the LLM to explore various objects, select the relevant one, and then provide the final answer. Thus, authors' approach seems to undermine the purpose of an "instruction." For instance, for the ambiguous prompt `white bear turned slightly`, a more suitable instruction with ambiguity removed would be, `identify the bear that is predominantly white with subtle color variations from white`.

Edit: The authors have sufficiently addressed my concerns and I'm increasing my rating.

**Questions:**

- For stronger VLMs (e.g., GPT-4o), I don't quite get the motivation for authors to adopt the "instantial experience" approach instead of the "empirical experience" approach. Intuitively, allowing VLMs to reason and explicitly output the clear instructions with ambiguities removed should almost always improve their performance, regardless of the ability of the VLMs themselves. I also didn't find an ablation study in the paper to support why we need the "instantial experience" approach for stronger VLMs.
- Additionally, per my comments in the review, I'd encourage authors design better prompts such that the generated instructions in their "empirical experience" approach do not defeat the purpose of an actual "instruction".

---

> ### Author Response · Authors · 2024-11-20
> **Authors Response 1/2**
>
> We appreciate your constructive advice!
>
> ## Q1: Trend of Performance Improvement
> In the previous version of Table 6 of the paper, we reported an extreme sample on the general dataset to show that even if there are slight fluctuations in the reasoning process, we can still return to an effective reasoning process. Additionally, since the general dataset contains mostly unambiguous instructions, the performance improvement is not as large as that of the ambiguous dataset.
>
> To further demonstrate the stability of our method, we extend the experiment to both ambiguous and general datasets, conducting $10$ experiments with different random seeds. We fixed the random seeds to $40$, $41$, $42$, $43$, $44$, $45$, $46$, $47$, $48$, and $49$. As shown in the tables below, we report the average accuracy of all experiments, as well as their maximum, minimum, and variance.
>
> | Budget | Avg    | Max   | Min   | Var   |
> | ----   | ------ |------ |------ |------ |
> | 1      | 55.38  |	56.22 |	54.10 |	0.45  |
> | 2      | 56.31  |	56.92 |	54.16 |	0.58  |
> | 3      | 57.58  |	58.56 |	56.76 |	0.31  |
> | 4      | 55.42  |	56.34 |	54.36 |	0.34  |
> | 5      | 56.17  |	57.44 |	55.14 |	0.46  |
>
> | Budget | Avg    | Max   | Min   | Var   |
> | ----   | ------ |------ |------ |------ |
> | 1      | 17.83  |	19.73 |	16.76 |	0.96  |
> | 2      | 19.92  |	21.89 |	18.77 |	0.95  |
> | 3      | 22.13  |	24.52 |	20.91 |	0.78  |
> | 4      | 18.82  |	20.54 |	17.30 |	0.84  |
> | 5      | 19.85  |	21.35 |	17.94 |	0.95  |
>
> This demonstrates that our method exhibits good stability on the ambiguous dataset and maintains a positive effect on the general dataset.
>
> The performance decrease observed when the budget $N > 3$ is due to the length of empirical experience prompt becoming longer, making it difficult for the current general-intelligent model to perform image-text understanding effectively. However, this is not an unsolvable problem. By adding an extra step where the model summarizes and simplifies long prompts, the prompt length can be shortened, and we find that the accuracy further improves when the budget $N > 3$:
>
> | Budget | Mean   | Max   | Min   | Div   |
> | ----   | ------ |------ |------ |------ |
> | 1      | 55.38  |	56.22 |	54.10 |	0.45  |
> | 2      | 56.31  |	56.92 |	54.16 |	0.58  |
> | 3      | 57.58  |	58.56 |	56.76 |	0.31  |
> | 4 + Summarization | 57.60  |	58.57 |	56.95 |	0.25  |
> | 5 + Summarization | 57.79  |	58.62 |	56.84 |	0.28  |
>
> | Budget | Mean   | Max   | Min   | Div   |
> | ----   | ------ |------ |------ |------ |
> | 1      | 17.83  |	19.73 |	16.76 |	0.96  |
> | 2      | 19.92  |	21.89 |	18.77 |	0.95  |
> | 3      | 22.13  |	24.52 |	20.91 |	0.78  |
> | 4 + Summarization | 22.90  |	25.13 |	21.43 |	0.88  |
> | 5 + Summarization | 23.00  |	25.28 |	21.78 |	0.69  |
>
> This means that if the model understands long texts well, our method can achieve increasingly higher results as the budget increases. If the model can only understand shorter texts, a budget of $3$ is sufficient to achieve good results.
>
> ## Q2: Response After Selecting Relevant Instruction
>
> We strongly agree that wrong instruction will affect the answer. Like humans, the model cannot guarantee that clear instructions have the same purpose as the original ambiguous one. Therefore, in our paper, we propose the response synthesis, which combines the original instruction and the clear instruction for the model's reference (see lines 245-250 of the paper), to address this issue.
>
> As shown in Table 5 of the paper, response synthesis can effectively mitigate the occurrence of errors in clear instructions.
>
> Inspired by your advice, we also explored a more proactive way to retain the purpose of the original instruction. Please refer to Q4.
>
> (Continued in Authors Response 2/2)

---

> ### Author Response · Authors · 2024-11-20
> **Authors Response 2/2**
>
> (From the Authors Response 1/2)
>
> ## Q3: Motivation of Instantial Experience
>
> We use instantial experience on the high-intelligent model because it has a faster reasoning speed. It only requires one time of API calling as all reasoning and reflection steps are jointly generated in one response. Plus, it does not need one-time optimization.
>
> Moreover, the high-intelligent model also supports empirical experience. In the Visual-O1$^−$ (SoM) line of Table 3 of the paper, empirical experience is also effective for it. However, due to the lack of specific reasoning for each sample, we observe a slight performance decrease compared to the instantial experience.
>
> Inspired by your comments, we slightly adjust the prompt to allow the reasoning process of instantial experience to generate clear instructions explicitly. We find that implementing this is very simple as it only needs to add a sentence to $p_{\mathrm{syn}}$: *Explicitly generate clear instructions within \<clear instruction> and \</clear instruction> tags before answering.* As shown in the table below:
>
> | Setting                                 | gIOU   |  cIoU  |
> | --------------------------------------- | ------ | ------ |
> | Empirical                               | 0.1143 | 0.1530 |
> | Instantial                              | 0.1304 | 0.1756 |
> | Instantial + Explicit Clear Instruction | 0.1316 | 0.1864 |
>
> We can see that explicitly generating clear instructions can further enhance the performance. We explain this as explicit clear instruction will help the model to review and better understand the instruction.
>
> ## Q4: Better Prompt to Keep the Original Purpose
>
> We greatly appreciate your suggestion! We further explore the idea you mentioned. We expand the functionality of response synthesis: instead of directly integrating the clear instruction with the original instruction, we designed a simple prompt to rewrite the clear instruction into a joint instruction guided by the purpose of the original instruction, ensuring keeping the original purpose while integrating as much information from the clear instruction as possible. We are surprised to find that the accuracy further increases:
>
> | Setting                      | gIOU   |  cIoU  |
> | ---------------------------- | ------ | ------ |
> | w/o Joint Instruction        | 0.1131 | 0.1215 |
> | w/ Joint Instruction         | 0.1245 | 0.1463 |
>
> We explain this as the extra step can keep the original purpose by fixing some misalignment between clear and original instructions. This also aligns with the findings in our paper that the response synthesis is important for Visual-O1.

---

> ### Comment · Reviewer_SvAq · 2024-11-20
>
> Thanks authors for the response! The significance test results, the summarization method under large budgets, and the Instantial + Explicit Clear Instruction method have significantly improved the paper, so I'm increasing the rating.

---

> > ### Author Response · Authors · 2024-11-21
> > **Thank You!**
> >
> > We are very appreciated for your thorough review and helping the paper into a better shape!
> >
> > Thank you again!
> >
> > Authors

---

### Author Response · Authors · 2024-11-20
**General Response**

We thank all reviewers for your patience and invaluable reviews.

We appreciate all positive feedback on our work: **important and interesting direction** (SvAq: "*an important research problem*"; FebY: "*an interesting direction*"); **role in complex samples** (FebY: "*... needs more on the harder case*"); **promising and adaptable approach** (SvAq: "*promising way to improve accuracy*", B9dk: "*adaptability indicates ... for a wider range of applications and users*"); **well-written and methodically clear** (SvAq: "*overall well-written, with clear methodology*"); **broad experiments and sufficient evidence** (SvAq: "*extensive experiments*", B9dk: "*extensive experimental results ... provide thorough evidence of the framework's effectiveness*"); **limited overhead in training and evaluation** (FebY: "*additional overhead ... is limited for both training and evaluation*").

We also appreciate all suggestions for our work, which will help our paper get in better shape. For specific responses to each question, please refer to the official comments posted below your review. We also revised our paper according to the suggestions and added additional experiments, examples, and analyses.

We are looking forward to further discussions with you!

---

### Meta-Review · Area_Chair_DBay · 2024-12-17

**Metareview:**

The paper was rated positively by the reviewers (8, 6, 6). They mentioned several strengths, such as addressing an important research problem, being well-written, scalable across different levels of model intelligence, and including extensive experiments. The provided rebuttal adequately addressed the concerns (details below). The AC checked the paper, the reviews, and the rebuttal, and follows the acceptance recommendation of the reviewers.

**Additional Comments On Reviewer Discussion:**

Reviewer SvAq had concerns such as lack of statistical significance and instructions directly answering the questions. Reviewer B9dk wanted to see an analysis of the failure examples and was concerned with minor improvements. Reviewer FebY also has concerns such as marginal improvements and confusing budget curves. The rebuttal addressed the comments well, which led to an increase of the scores by two reviewers.

---

### Decision · Program_Chairs · 2025-01-22

Accept (Poster)